# Spatiotemporal imaging and shaping of electron wave functions using novel attoclock interferometry

Peipei Ge[1,2], Yankun Dou[1], Meng Han [3], Yiqi Fang[1], Yongkai Deng[1], Chengyin Wu[1], Qihuang Gong [1,4,5] & Yunquan Liu [1,4,5] ✉

Electrons detached from atoms by photoionization carry valuable information about light-atom interactions. Characterizing and shaping the electron wave function on its natural timescale is of paramount importance for understanding and controlling ultrafast electron dynamics in atoms, molecules and condensed matter. Here we propose a novel attoclock interferometry to shape and image the electron wave function in atomic photoionization. Using a combination of a strong circularly polarized second harmonic and a weak linearly polarized fundamental field, we spatiotemporally modulate the atomic potential barrier and shape the electron wave functions, which are mapped into a temporal interferometry. By analyzing the two-color phase-resolved and angle-resolved photoelectron interference, we are able to reconstruct the spatiotemporal evolution of the shaping on the amplitude and phase of electron wave function in momentum space within the optical cycle, from which we identify the quantum nature of strong-field ionization and reveal the effect of the spatiotemporal properties of atomic potential on the departing electron. This study provides a new approach for spatiotemporal shaping and imaging of electron wave function in intense light-matter interactions and holds great potential for resolving ultrafast electronic dynamics in molecules, solids, and liquids.

Strong-field ionization serves as a cornerstone for various ultrafast phenomena such as high harmonic generation[1,2], laser-induced electron diffraction[3,4], nonsequential double ionization[5,6] and photoelectron holography[7–9] that underpin strong-field physics and attosecond science[10–12]. The liberated photoelectrons carry dynamical information about the ionization process[13–15] and encode structural details of atoms and molecules[16–20]. To reveal the underlying ionization dynamics, it is essential to track and probe the electron dynamics on its natural timescale ($1as = 10^{-18}s$), which is the primary goal of attosecond

science. Recent progress in attosecond metrology has opened up the possibilities both for the characterization of electron wave function and for its laser-driven shaping in real time[21–32]. Notable techniques include attosecond streak camera[21,22], attosecond transient absorption spectroscopy[23,24], high harmonic spectroscopy[25,26], and attoclock[27]. Among these techniques, attoclock serves as a powerful tool in time-resolving electron dynamics in strong-field regime. It achieves attosecond precision based on the fact that in strong-field ionization by circularly polarized laser pulses, the ionization time of photoelectron

[1]State Key Laboratory for Mesoscopic Physics and Frontiers Science Center for Nano-optoelectronics, School of Physics, Peking University, Beijing 100871, China. [2]Wuhan National Laboratory for Optoelectronics and School of Physics, Huazhong University of Science and Technology, Wuhan 430074, China. [3]J. R. Macdonald Laboratory, Department of Physics, Kansas State University, Manhattan, KS 66506, USA. [4]Collaborative Innovation Center of Extreme Optics, Shanxi University, Taiyuan, Shanxi 030006, China. [5]Peking University Yangtze Delta Institute of Optoelectronics, Nantong 226010 Jiangsu, China. ✉e-mail: yunquan.liu@pku.edu.cn

is mapped to the emission angle. Since the pioneering implement[27], attoclock has been widely used to probe the tunneling time delay[28,29], reveal the tunneling geometry[30], time the release of electrons[31], and image the ultrafast electronic and nuclear motion inside molecules[32], providing insight into the fundamental aspects of intense light-matter interactions.

However, in conventional attoclock experiments, owing to the adoption of few-cycle laser pulses, the photoelectron interference effect has been largely suppressed. As a result, the phase information of ionized electrons is lost. This hinders the complete characterization of electron wave function, especially for the unraveling of the quantum properties embedded in strong-field ionization. Generally, the electron dynamics in strong-field ionization involves passing through the potential barrier suppressed by strong laser fields and propagating in the continuum states. While the latter can be well understood within the classical mechanics, the former dynamics in the classically forbidden region, which dictates the main quantum nature of strong-field interaction and influences the following dynamics, remains obscure. Although recent advances in experimental techniques have offered a glimpse of the under-barrier dynamics[26,33–36], comprehensively understanding it is still far off. This calls for a detailed look into the barrier. In particular, two fundamental questions, i.e., how does the field-modulated potential barrier influence the ionization dynamics and can we resolve the fingerprint of modulated potential barrier from the ionized electron wave function, need to be addressed. Answering these questions requires not only an accurate modulation of the barrier but also the measurement and shaping on the amplitude and phase of electron wave function.

In this work, we demonstrate a novel two-color attoclock technique that integrates with a temporal interferometry to shape and image electron wave function during strong-field ionization and thereby to uncover the under-barrier dynamics. We achieve this by using a combination of a strong second harmonic field with circular polarization at 400 nm and a weak linearly polarized fundamental field at 800 nm. Here, the fundamental field introduces a spatiotemporal perturbation to the potential barrier bended by the intense second harmonic that depends on the two-color relative phase and the spatial orientation of the potential barrier. This perturbation shapes the amplitude and phase of the ionized electron wave function and is finally mapped into the temporal interferometry. Through the analysis of the two-color phase-resolved photoelectron interference at specific emission angles, we fully reconstruct the temporal shaping of electron wave function, thereby visualizing how the ionization dynamics evolves with respect to a fast-oscillating potential barrier under different interaction configurations. Furthermore, based on the analysis of angle-resolved photoelectron interference at fixed two-color relative phases, we are able to take snapshots of the shaped electron wave function in the full momentum space and thus reveal the effect of the spatial property of potential barrier on electron dynamics. This innovative two-color attoclock interferometry provides valuable insights into the quantum dynamics in strong-field ionization dictated by the field-modulated potential barrier. It has promising implications for visualizing ultrafast electronic dynamics inside molecules, solids, and liquids in time and space.

## Results and discussion
### Two-color attoclock interferometry
In our experiment, a strong circularly polarized field at 400 nm (3k Hz, 40 fs) with an intensity of $1.42 \times 10^{14}$ W/cm$^2$ is employed to ionize Ar atoms. Here, the polarization plane of 400 nm field is in $x–z$ plane. As illustrated in Fig. 1a, the rotating electric field vector of 400 nm bends the atomic potential and forms a rotating Coulomb barrier, through which the bound electron is released. Depending on their ionization times, the emitted electrons would be deflected to different emission angles. Here, the emission angle $\theta$ is defined as the angle between the

direction of the electron final momentum $\mathbf{p}$ and $+z$ axis. This time-to-angle mapping is the principle of attoclock. Since multicycle pulses have been used, the electrons that release at times separated by laser cycles would interfere with each other at a specific angle, giving rise to the intercycle interference[37], which manifests as the well-known above-threshold ionization (ATI)[38] peaks in photoelectron momentum (energy) spectrum. This defines the angle-resolved temporal interferometry in attoclock geometry. Experimentally, the three-dimensional momentum distributions (PMDs) of photoelectrons are measured by cold-target recoil-ion reaction momentum spectroscopy (COLTRIMS)[39]. (Details of the experimental setup can be found in "Methods"). Since the circular field vector is angularly uniform, the measured PMD on the $x–z$ polarization plane, as shown in Fig. 1b, reveals a series of isotropic ATI rings.

We then add a weak linearly polarized field at 800 nm with the intensity of $8.8 \times 10^{11}$ W/cm$^2$ and its polarization along $z$ direction to manipulate the spatiotemporal shape of the potential barrier and introduce changes in the attoclock interferometry constructed by 400 nm circular fields. As depicted in Fig. 1c, in two-field fields with a fixed relative phase $\varphi$, both the polarization configuration (indicated by the blue and red arrows) and the perturbative field strength are changing over time. The spatially rotating potential barrier undergoes spatial modulations imposed by the perturbative field that depends on the ionization time. Consequently, the electrons that release at different times would experience distinct modulations. In particular, for the two electron wave packets that are released at times separated by one 400 nm cycle, they experience opposite modulations as the potential barrier is perturbed by opposite 800 nm electric field vector (the modified potential barriers along $z$ direction are depicted by the red and blue dashed lines in the lower panel of Fig. 1c). Such modulation would be finally imprinted into the angle-resolved photoelectron interference pattern. And for each emission angle $\theta$, it represents a distinct interaction configuration. Therefore, by analyzing the angle-resolved photoelectron interference pattern and retrieving the amplitude and phase of electron wave function for each emission angle, one can realize the spatial imaging of the ionization dynamics under various interaction configurations in the full momentum space. Besides, scanning the two-color relative phase also provides an additional knob to manipulate the potential barrier in time domain. As shown in Fig. 1c, varying the two-color relative phase, the perturbative field strength is modified for ionization at different times (angles). By inspecting the two-color phase-dependent photoelectron interference at a specific emission angle, one is allowed to monitor how the ionized electron wave function evolves with respect to the fast-oscillating potential barrier, thus achieving the temporal imaging of the ionization dynamics.

Figure 1d displays the measured two-color phase-integrated PMD on the $x–z$ polarization plane. One can see that adding a weak fundamental field gives rise to the formation of sideband structures between adjacent ATI rings on PMD. Moreover, due to the utilization of linearly polarized perturbative fields, the two-color phase-integrated interference pattern becomes no longer isotropic. Instead, it shows a slight angle-dependence. To visualize the spatiotemporal modulations induced by the perturbative field in the attoclock interferometry, we inspect the measured two-color phase-resolved photoelectron energy spectra at different emission angles. The corresponding spectra for $\theta = 0°$, $45°$, $90°$, and $135°$ are displayed in Fig. 2a–d, respectively. Apparently, the spectra are dominated by equidistant interference peaks spaced by the 800 nm photon energy, which correspond to the alternating ATI peaks and sidebands. This is analogous to the traditional optical interference fringes in real space. More importantly, the photoelectron interference pattern undergoes very different evolution with respect to the relative phase for different emission angles, implying the time- and space-dependent features of the ionization dynamics in two-color attoclock geometry.

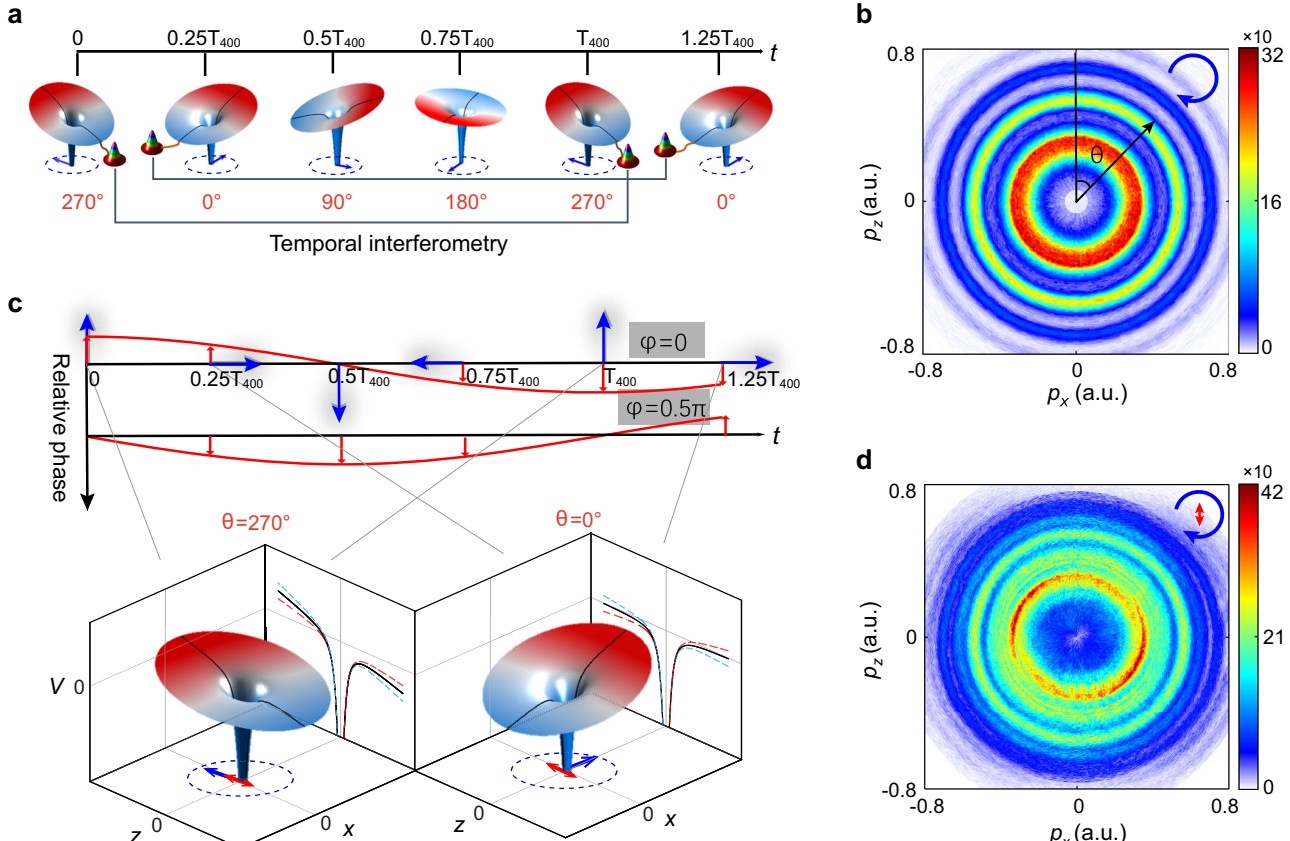

**Fig. 1 | Schematic illustration of two-color attoclock interferometry. a** Common attoclock geometry constructed by multicycle 400 nm circular fields. The time-dependent rotating barriers formed by 400 nm circular fields are depicted. The time-to-angle mapping is given. **b** Measured PMD of strong-field ionization of Ar atoms on x–z polarization plane in 400 nm attoclock geometry. The emission angle θ is defined as the angle between the direction of the final momentum **p** and +z axis. **c** Temporal evolution of the two-color field configuration in the novel attoclock geometry, where a weak linearly polarized field at 800 nm along z direction is added to perturb the attoclock established by 400 nm circular fields. For times separated by one 400 nm optical cycle $T_{400}$, the perturbative 800 nm field points to opposite direction, thus leading to opposite modulation on the barrier, as

depicted in the lower panel. Here, the barriers at times of $t = 0$ ($T_{400}$) and $t = 0.25T_{400}$ ($1.25T_{400}$) which correspond to the situations of $\theta = 270°$ and $\theta = 0°$, respectively, are exemplified. They are manipulated by the fundamental field in parallel and perpendicular directions, respectively. The black lines show the unperturbed potential curves along z direction, while the red and blue dashed lines represent the oppositely modulated potential curves by the fundamental field at times separated by one 400 nm cycle. Varying the relative phase between two colors, the polarization configuration remains unchanged, whereas the perturbative field strength at the ionization time changes. **d** Measured two-color phase-integrated PMD on x–z polarization plane in two-color attoclock geometry.

## SFA analysis of photoelectron interference

To understand the two-color phase- and angle-dependent photoelectron interference in the geometry, we resort to the strong-field approximation (SFA)[40]. This model has been verified to work well in describing strong-field ionization by sculptured circular fields[36,41,42]. Within saddle-point approach[43], the transition amplitude of the photoelectron, from the initial ground state $|\psi_0\rangle$ to the final Volkov state $|\psi_{\mathbf{p}}^V\rangle$ with the final momentum **p**, can be regarded as the coherent supposition of all quantum orbits, i.e., $M_{\mathbf{p}} \sim \sum_i \rho_s(\mathbf{p})e^{iS(\mathbf{p},t_s^{(i)})}$. Here, $\rho_s(\mathbf{p}) \sim \langle \mathbf{p} + \mathbf{A}(t_s)|\mathbf{r} \cdot \mathbf{E}(t_s)|\psi_0(\mathbf{r})\rangle$ is the pre-exponential factor with $\mathbf{A}(t)$ and $\mathbf{E}(t)$ denoting the vector potential and electric field of laser pulses, respectively. The saddle-point $t_s$, which can be derived from the equation of $[\mathbf{p} + \mathbf{A}(t_s)]^2/2 + I_p = 0$ with $I_p$ being the ionization potential, represents the complex ionization instant of the quantum orbit. $S(\mathbf{p},t_s) = -\int_{t_s}^{\infty}[\mathbf{p} + \mathbf{A}(t)]^2/2dt + I_p t_s$ is the classical action (or complex phase) of the quantum orbit. In single 400 nm circular fields, the photoelectron interference at the final momentum **p** along θ originates from the intercycle interference among quantum orbits whose ionization instants are separated by 400 nm cycles. When adding a weak linearly polarized 800 nm field, as illustrated in Fig. 1c, the quantum orbits that release in adjacent 400 nm cycles with their ionization instants satisfying $t_s^{(2)} = t_s^{(1)} + T_{400}$ would experience opposite

modulations by the perturbative fundamental field. Here, $T_{400}$ denotes the optical period of 400 nm laser pulses. In the next 800 nm cycle, there arises a similar pair of quantum orbits with their ionization instants satisfying $t_s^{(3)} = t_s^{(1)} + T_{800}$ and $t_s^{(4)} = t_s^{(2)} + T_{800}$ ($T_{800}$ represents the optical period of 800 nm pulses). Considering the periodicity of the light field, four quantum orbits releasing in two 800 nm cycles can well account for the photoelectron interference in two-color synthesized fields. In this case, the photoelectron interference in two-color fields at the emission angle θ can be written as

$$I(\mathbf{p},\varphi) = |\psi_1 + \psi_2 + \psi_3 + \psi_4|^2 = |\psi_1 + \psi_2|^2|1 + e^{ib}|^2 \quad (1)$$

where $\psi = \rho(\mathbf{p})e^{iS}$ represents the ionized electron wave packet, $b = (U_p^{(400)} + U_p^{(800)} + E_k + I_p)T_{800}$ is the action difference due to the time difference of traveling in the continuum. $U_p$ is the pondermotive energy and $E_k = p^2/2$ denotes the photoelectron energy. And $\psi_3 = \psi_1 e^{ib}$, $\psi_4 = \psi_2 e^{ib}$, respectively.

Figure 2e–h displays the calculated photoelectron energy spectra as a function of the two-color relative phase based on Eq. (1) for the selected angles. They show good agreement with the measured results. As for the slight discrepancies in low-energy region, this can be attributed to the neglect of Coulomb effect in SFA model (Comparison

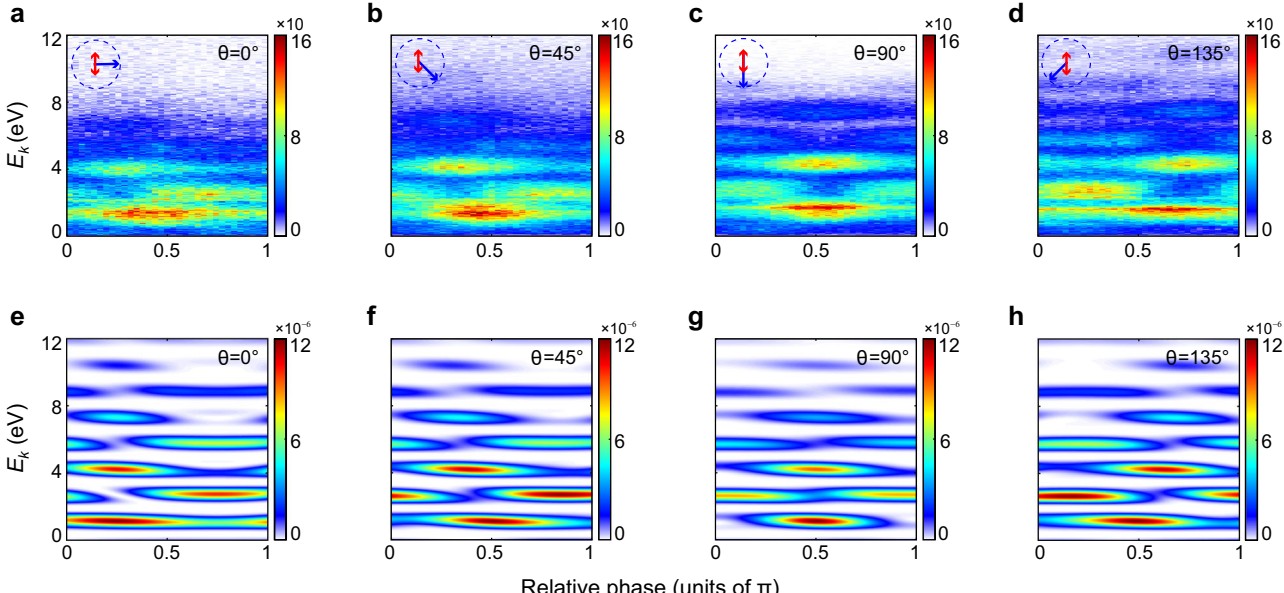

**Fig. 2 | Two-color phase-resolved photoelectron energy spectra. a–d** Measured two-color phase-resolved photoelectron energy spectra at the emission angles of $\theta = 0°$, 45°, 90° and 135°, respectively. **e–h** Calculated two-color phase-resolved energy spectra by SFA model at the emission angles of $\theta = 0°$, 45°, 90° and 135°, respectively. The blue arrow on the top denotes the direction of the rotating 400 nm field vector. The red arrow indicates the polarization direction of 800 nm linearly polarized field.

among the experimental results, SFA and Coulomb-corrected SFA (CCSFA) calculations can be seen in Supplementary Fig. 2). According to the time-to-angle mapping governed by SFA, i.e., $\mathbf{p} = -\mathbf{A}(t_0)$, we can determine the direction of the 400 nm electric field vector at the ionization instant $t_0$ for these emission angles, and thus obtain the corresponding polarization configurations, as labeled by the blue and red arrows in Fig. 2a–d. Note that since the 800 nm field is perturbative weak, we have neglected its influence on the time-to-angle mapping.

Having demonstrated good agreement between the experiment and SFA calculations, we continue analyzing the photoelectron interference using the SFA model. In two-color synthesized fields, the complex phase of the electron wave packet can be expressed as $S(\mathbf{p},t_s) = -\int_{t_s}^{\infty}[\mathbf{p} + \mathbf{A}_{400}(t) + \mathbf{A}_{800}(t,\varphi)]^2/2dt + I_p t_s = S_0 + \sigma - \int_{t_s}^{\infty}[\mathbf{A}_{800}(t,\varphi)]^2/2dt$, where $S_0(\mathbf{p},t_s) = -\int_{t_s}^{\infty}[\mathbf{p} + \mathbf{A}_{400}(t)]^2/2dt + I_p t_s$ represents the action induced solely by the 400 nm circular field, $\sigma = -\int_{t_s}^{\infty}[\mathbf{p} + \mathbf{A}_{400}(t)] \cdot \mathbf{A}_{800}(t,\varphi)dt$ is the additional action induced by the weak linearly polarized 800 nm field and $-\int_{t_s}^{\infty}[\mathbf{A}_{800}(t,\varphi)]^2/2dt$ is the high-order quantity and can be reduced to a linear phase shift $U_p^{(800)}t_s$. Consequently, the photoelectron interference in two-color fields at the emission angle $\theta$ in the energy space can be rearranged as

$$
\begin{aligned}
I(E_k,\varphi) &= |\psi_1 + \psi_2 + \psi_3 + \psi_4|^2 \\
&= 2W_0^2(e^{2\text{Im}[\sigma]} + e^{-2\text{Im}[\sigma]})[1 + \cos(2E_k T_{400} + 2a)] \\
&\quad + 4W_0^2 \cos(E_k T_{400} + 2\text{Re}[\sigma] + a) \\
&\quad + 2W_0^2 \cos(E_k T_{400} - 2\text{Re}[\sigma] + a) \\
&\quad + 2W_0^2 \cos(3E_k T_{400} + 2\text{Re}[\sigma] + 3a)
\end{aligned}
\tag{2}
$$

Here, $W_0 = \rho(\mathbf{p})e^{-\text{Im}[S_0]}$ represents the amplitude of ionized electron wave packet in 400 nm circular field. $\text{Re}[\sigma]$ and $\text{Im}[\sigma]$, which refer to the real part and imaginary part of the additional phase $\sigma$, respectively, quantify the phase and amplitude modulations of the electron wave function induced by 800 nm perturbative field. Note that the amplitude modulation induced by the change of $\rho(\mathbf{p})$ when adding a weak 800 nm field has been neglected since it is much smaller as compared with that induced by the change of imaginary part of the complex phase (Supplementary Fig. 3). The $a = (I_p + U_p^{(800)} + U_p^{(400)})T_{400}$

denotes a constant phase. Detailed derivation of Eq. (2) can be found in Supplementary Information. When the perturbative field is absent, the two-color interference will be reduced to the single-color case with the formula governed by

$$
\begin{aligned}
I(E_k,\varphi) &= |\psi_1 + \psi_2 + \psi_3 + \psi_4|^2 \\
&= 4W_0^2[1 + \cos(2E_k T_{400} + 2a)] \\
&\quad + 6W_0^2 \cos(E_k T_{400} + a) \\
&\quad + 2W_0^2 \cos(3E_k T_{400} + 3a)
\end{aligned}
\tag{3}
$$

From the expressions in Eqs. (2) and (3), one can see that the photoelectron interference among four electron wave packets emitting within two 800 nm cycles contains multiple frequencies along $E_k$, i.e., the zero-frequency component, $1f_x$, $2f_x$, and $3f_x$ frequency components with $f_x = T_{400}/2\pi$. This indicates that we can apply the Fourier transform analysis[44], which has been frequently used in analyzing the traditional optical interferogram, to the photoelectron interference in order to extract the amplitude and phase modulations of the electron wave function induced by the spatiotemporal manipulation of potential barrier.

## Fourier transform analysis for fully retrieving the electron wave function

As presented in Eq. (2), the amplitude modulation $\text{Im}[\sigma]$ caused by the perturbative field is encoded in the amplitudes of the zero-frequency term and $2f_x$ frequency term of the photoelectron interference. Here, we utilize the zero-frequency term of the photoelectron interference to extract $\text{Im}[\sigma]$. By performing Fourier transform on the photoelectron interference along $E_k$ at a specific angle $\theta$, we obtain the frequency spectra in single- and two-color cases. Then, we filter out the zero-frequency components and perform the inverse Fourier transform. Accordingly, we obtain the amplitudes of zero-frequency terms in single- and two-color fields, which correspond to $4W_0^2$ and $2W_0^2(e^{2\text{Im}[\sigma]} + e^{-2\text{Im}[\sigma]})$, respectively. Dividing $2W_0^2(e^{2\text{Im}[\sigma]} + e^{-2\text{Im}[\sigma]})$ by $4W_0^2$, we obtain the ratio $k = (e^{2\text{Im}[\sigma]} + e^{-2\text{Im}[\sigma]})/2$, from which $\text{Im}[\sigma]$ can be analytically derived with the formula of $\text{Im}[\sigma] = 0.5ln(k + \sqrt{k^2 - 1})$ (The

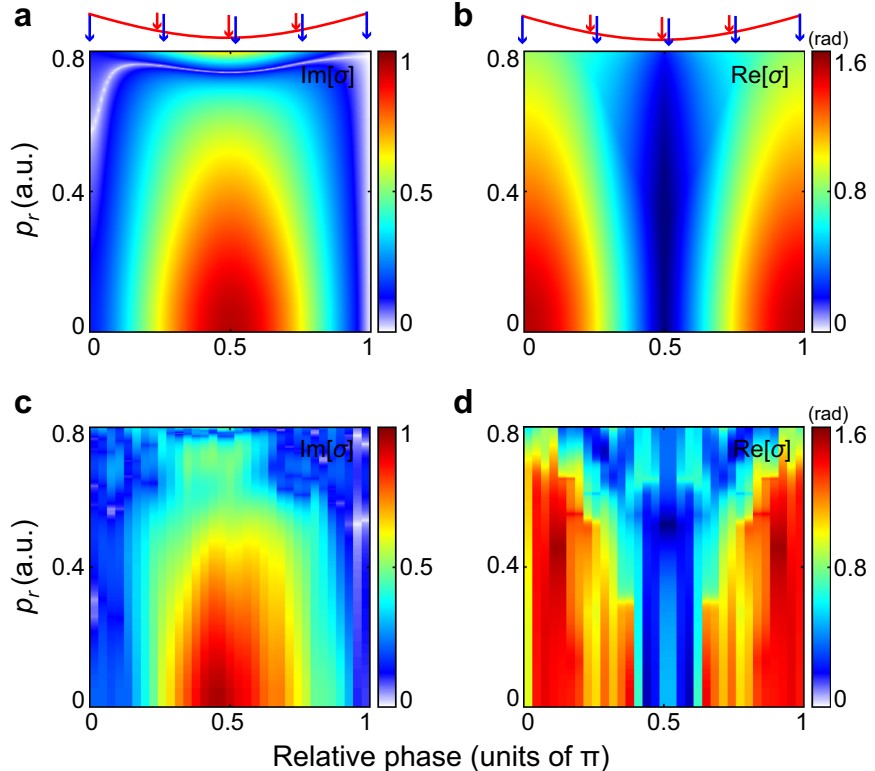

**Fig. 3 | Temporal shaping of ionized electron wave function in parallel inter-action configuration. a, b** Retrieved two-color phase-dependent Im[$\sigma$] and Re[$\sigma$] from the SFA calculated phase-dependent photoelectron energy spectrum (as shown in Fig. 2g). The two-color field configurations are shown on the top, with the blue and red arrows indicating the electric field vectors of 400 nm and 800 nm pulses, respectively. The red lines denote the oscillating 800 nm perturbative field. **c, d** Same as (**a**), (**b**) but retrieved from the measured result as shown in Fig. 2c.

other root equaling $0.5ln(k - \sqrt{k^2 - 1})$ is unphysical and should be discarded). While to retrieve the phase modulation Re[$\sigma$], we focus on the $3f_x$ frequency term, i.e., $2W_0^2\cos(3E_kT_{400} + 2\text{Re}[\sigma] + 3a)$, as Re[$\sigma$] is directly imprinted into its phase. By applying the Fourier transform to the two-color phase-dependent photoelectron interference along $E_k$, we obtain the two-color phase-resolved frequency spectrum, from which we filter out the $3f_x$ frequency term and shift it to the zero-frequency position. We then perform the inverse Fourier transform and obtain a complex value of $2W_0^2e^{i(2\text{Re}[\sigma]+3a)}$ whose phase corresponds to 2Re[$\sigma$]+3a. Afterward, we subtract the constant phase 3a from the retrieved phase and finally access the phase modulation Re[$\sigma$]. In fact, within the saddle-point approach, the amplitude and phase modula-tions can be analytically derived by substituting the expression of two-color electric field into the definition of $\sigma$. The analytical formulas for Im[$\sigma$] and Re[$\sigma$] are given in Eq. (4) in "Methods". By comparing the calculated Im[$\sigma$] and Re[$\sigma$] (Supplementary Fig. 4) with the retrieved results, one can validate the Fourier transform analysis.

**Temporal shaping and imaging of electron wave function**
In the following, we apply the Fourier transform analysis to the two-color phase-dependent photoelectron interference at specific angles in order to retrieve the temporal evolution of the ionized electron wave function through a fast-oscillating potential barrier. Here, we concentrate on two intriguing strong-field interaction configurations, in which the potential barrier is manipulated by the weak fundamental field along longitudinal (parallel) and transverse (perpendicular) directions, corresponding to the situations at $\theta = 90°$ and $0°$, respectively.

In Fig. 3, we present the retrieved Im[$\sigma$] and Re[$\sigma$] spectra from the SFA calculations and from the measured results in the parallel con-figuration ($\theta = 90°$). Good agreement has been achieved between the experiment and the SFA calculations. As depicted, in this configura-tion, the ionizing electric field vector points to the negative z axis, parallel with the perturbative field. Varying the two-color relative phase modifies the perturbative field strength at the ionization instant. Interestingly, as the perturbative field increases from zero to its max-imum, the amplitude modulation Im[$\sigma$] on electron wave function increases from zero to maximum, whereas the phase modulation Re[$\sigma$] decreases from maximum to zero.

In parallel interaction configuration, increasing the perturbative field strength is equivalent to reducing the thickness of the potential barrier along the ionization direction. More photoelectrons would be ionized through a narrower barrier, resulting in the increased ioniza-tion amplitude. Since the classical propagation after ionization does not influence the ionization probability[33], the retrieved Im[$\sigma$] directly reflects changes in the imaginary phase accumulated during the elec-trons' under-barrier motion. On the other side, the photoelectron acquires an additional drift momentum along z direction governed by $-\mathbf{A}_{800}(t,\varphi)$ from the perturbative field. With the increase of the per-turbative electric field, the drift momentum decreases. Such momen-tum is parallel to the ionizing direction but perpendicular to the final momentum $\mathbf{p}$ of electron. It would induce an additional real phase during the under-barrier excursion and also a lateral phase shift during the classical propagation (Supplementary Fig. 5a, b). These two phases jointly contribute to the retrieved phase modulation Re[$\sigma$] as shown in Fig. 3b, d. Besides, one could see that the retrieved Im[$\sigma$] and Re[$\sigma$] spectra exhibit prominent energy-dependences, i.e., the photoelec-trons with lower energies (or lower radial momenta $p_r$) experience larger modulations than those of higher energies. This phenomenon can be ascribed to the energy-dependent imaginary part $t_i$ of the ionization time $t_s$ in circular fields, as shown in Supplementary Fig. 1, which quantifies the excursion time under the barrier and determines

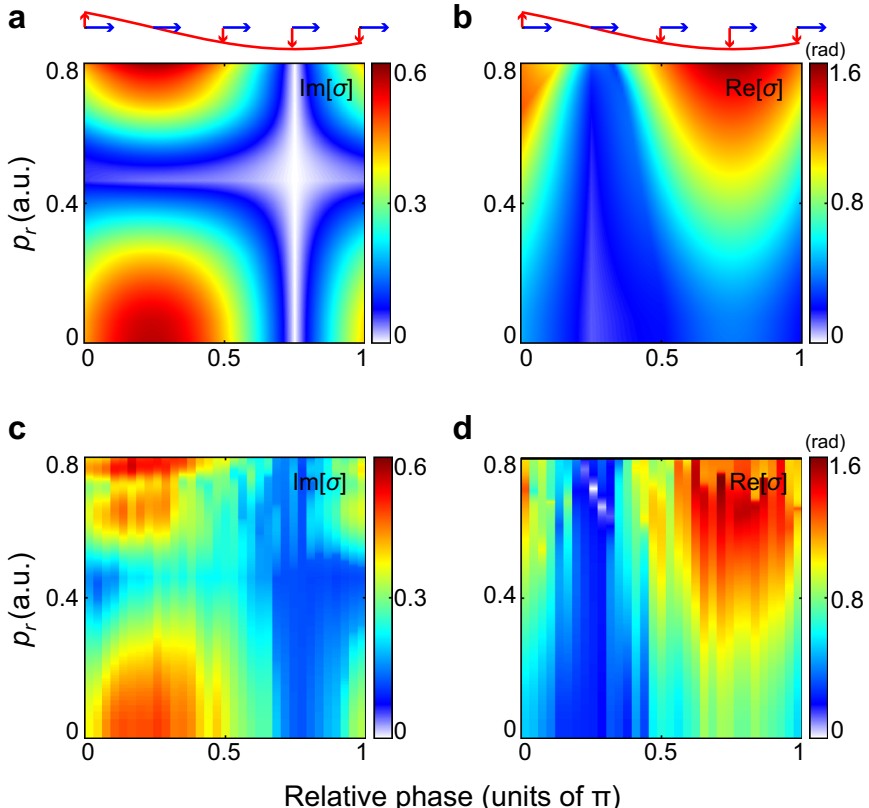

**Fig. 4 | Temporal shaping of ionized electron wave function in perpendicular interaction configuration. a, b** Retrieved two-color phase-dependent Im[$\sigma$] and Re[$\sigma$] from the SFA calculated phase-dependent photoelectron interference (as shown in Fig. 2e). The two-color field configurations are shown on the top, with the blue and red arrows indicating the electric field vectors of 400 nm and 800 nm pulses, respectively. The red lines depict the oscillating 800 nm perturbative field. **c, d** Same as (**a**), (**b**) but retrieved from the measured result as shown in Fig. 2a.

the real and imaginary phase of electron accumulated during under-barrier motion [see Eq. (4) in "Methods"].

Another intriguing scenario is that the potential barrier is manipulated transversely, corresponding to the case of $\theta = 0°$. Based on the results shown in Fig. 2a, e, we extract the phase-dependent amplitude and phase modulations of the electron wave function in the perpendicular configuration. The corresponding results are displayed in Fig. 4. Discrepancies in the high-energy part of the retrieved Im[$\sigma$] spectra between SFA calculation and experiment can be attributed to the imperfect filter of zero-frequency component from the Fourier frequency spectrum of the experimental result that hampers the accurate retrieval of Im[$\sigma$]. In this configuration, the ionizing 400 nm electric field vector points to +x direction and the potential barrier is perturbed transversely by the weak fundamental field. The retrieved Im[$\sigma$] and Re[$\sigma$] spectra reveal distinct phase- and energy-dependences in comparison to the case of parallel configuration. Specifically, when the perturbative field approaches zero, the amplitude modulation maximizes, whereas the phase modulation vanishes. Increasing the perturbative field to its maximum, the amplitude modulation gradually disappears, while the phase modulation reaches the maximum. Moreover, the two flanks of the electron wave packet, i.e., the lower- and higher-energy parts, seem to suffer a larger amplitude modulation than the central part which has a vanishing transverse momentum at the ionization instant and is finally streaked to $p_r$ ~ 0.4 a.u. by the 400 nm circular field. This implies that under the barrier, the imaginary phase of electrons with larger transverse momenta is more susceptible to be influenced by the perpendicular perturbative field. For the retrieved Re[$\sigma$] spectra, the electrons with higher energies experience a larger phase modulation than the low-energy electrons.

Note that different from the case in parallel configuration, the potential barrier in perpendicular configuration is deformed in transverse direction and its thickness along the ionization direction remains unchanged. Moreover, the additional momentum acquired by the photoelectron from the perturbative field is perpendicular to the ionization direction but parallel to the final momentum. When the perturbative electric field is vanishing ($\varphi = 0.25\pi$), the additional momentum maximizes. This causes the largest modulation on the imaginary phase of electrons. However, since the shape of potential barrier remains unchanged in this case, the real phase of the electrons accumulated during the under-barrier motion and the following propagation keeps unchanged (Supplementary Fig. 5c, d). Increasing the perturbative field to its maximum ($\varphi = 0.75\pi$), the momentum acquired by electrons along z axis vanishes, and, as a result, the modulation on the imaginary phase of electrons disappears. Nevertheless, it should be noted that the potential barrier in this case experiences the largest lateral perturbation. This results in the largest modulation on the real phase of the electron during its entire motion.

From the retrieved temporal evolutions of shaped electron wave functions in parallel and perpendicular configurations, we find that the employed field geometry also allows for selective manipulation of the amplitude and phase of electron wave function in strong-field ionization, simply by modifying the two-color relative phase. For example, in the parallel configuration, by adjusting the two-color relative phase to $\varphi = 0.5\pi$, one can solely manipulate the amplitude of electron wave function while keep its phase unchanged. When modifying $\varphi$ to 0, the selective shaping of the phase of electron wave function is enabled with its amplitude remaining unchanged. In principle, if performing Fourier transform analysis on the measured two-color phase-dependent photoelectron interference for each emission angle,

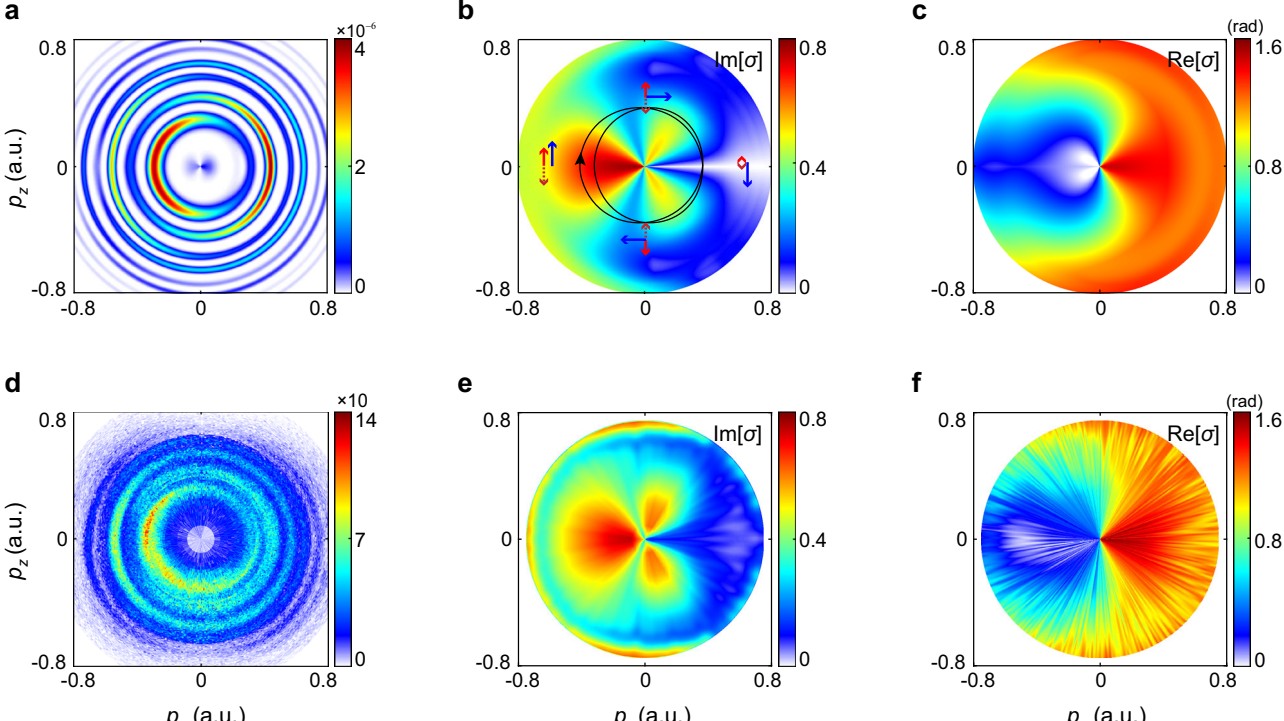

**Fig. 5 | Spatial imaging of the ionized electron wave function in momentum space under two-color synthesized fields. a** Calculated PMD in two-color synthesized field with the relative phase $\varphi = 0$ based on the SFA model. **b, c** Retrieved momentum-resolved Im[$\sigma$] and Re[$\sigma$] spectra from the SFA calculated PMD (shown in **a**). In (**b**), the two-color synthesized electric field has been plotted. Also, the corresponding field configurations at specific angles are labeled. **d** Measured PMD in the two-color synthesized field with the relative phase $\varphi = 0$. **e, f** Retrieved momentum-resolved Im[$\sigma$] and Re[$\sigma$] spectra from the measured PMD (shown in **d**).

one could access the complete information about electron wave function in time and space under this geometry, thereby shedding light into the underlying quantum dynamics of intense light-matter interactions.

## Spatial shaping and imaging of electron wave function

Apart from temporal shaping and imaging of electrons, spatial shaping and imaging also arouses great interest of scientists, since the spatial property of emitted electrons is closely tied to electronic structures and it dictates the subsequent electron and nuclear dynamics inside molecules such as electron localization[45] and charge migration[46]. As illustrated by Fig. 1c, in the two-color attoclock geometry, the photoelectron interference along different emission angles encodes ionization dynamics under different strong-field interaction configurations. This reflects the spatial property of the attoclock geometry. By analyzing the angle-resolved photoelectron interference at a fixed two-color relative phase, one is allowed to capture the snapshots of the shaped electron wave function in the full momentum space for diverse configurations.

Figure 5a and d display the calculated and measured PMDs in two-color fields with $\varphi = 0$, respectively. The SFA calculation basically reproduces the main features of the measured PMD, except for the low-energy part which suffers from the Coulomb effect. To facilitate analysis on photoelectron interference, we have transformed the PMD into the angle-resolved photoelectron energy spectrum. The spectrum is rotated by a specific angle in order to eliminate the Coulomb effect. Here, the rotating angle can be read from the comparison between the calculated angle-resolved spectra using the SFA and CCSFA models (Supplementary Fig. 6). Then, we apply the Fourier transform analysis to the angle-resolved photoelectron energy spectrum and retrieve the angle-resolved Im[$\sigma$] and Re[$\sigma$] spectra in energy space. These spectra are later transformed back to the momentum space.

As shown in Fig. 5b, e and Fig. 5c, f, the retrieved Im[$\sigma$] and Re[$\sigma$] spectra reveal obvious angle- and energy-dependent features. And they are symmetric about $p_x$ axis. Actually, such symmetry reflects the field geometry of the two-color synthesized fields, as depicted by the black line in Fig. 5b. Particularly, as the ionization amplitude critically depends on the electric field strength, the retrieved angle-dependent amplitude modulation directly maps the evolution of sculptured two-color laser fields. Scanning the emission angle, the spatially rotating potential barrier is manipulated along the $z$ direction by the fundamental field of different strength (see the labeled two-color field configurations in Fig. 5b). The emission angle indeed encodes the spatial property of the field-modulated potential barrier. Hence, the retrieved angle-resolved Im[$\sigma$] and Re[$\sigma$] spectra in the full momentum space reveals the impact of spatial property of potential barrier on the electron dynamics, demonstrating the spatial imaging capability of the two-color attoclock geometry. This capability provides unique possibilities to probe the electronic environment of molecules, solids, and liquids.

In summary, we have demonstrated a novel two-color attoclock interferometry that enables the spatiotemporal shaping and imaging of the electron wave function in strong-field ionization. This interferometry encompasses a wide range of strong-field interaction configurations, thereby allowing the scrutiny of the underlying ionization dynamics in diverse scenarios. Through the analysis of the two-color phase-dependent photoelectron interferences at specific angles, we have gained insights into the temporal evolution of the ionization dynamics with respect to a fast-oscillating potential barrier under different interaction configurations, especially for the quantum dynamics under the barrier. This field geometry, as demonstrated in our experiment, can also be exploited for selectively manipulating the amplitude and phase of the electron wave function. Besides that, the analysis of the angle-resolved photoelectron interference at a specific two-color relative phase permits the spatial imaging of the electron

dynamics in the full momentum space. Our study illuminates puzzles in understanding how the spatiotemporal properties of Coulomb barrier influence the ionization process and shape the electron wave function. Looking forward, this novel attoclock interferometry has important implications for spatiotemporally resolving ultrafast electronic dynamics inside molecules following photoionization such as charge migration[46] and charge transfer processes[47], paving the way to real-time measurement and control of chemical transformations.

## Methods

### Experimental setup

The second harmonic pulse at 400 nm is obtained by frequency doubling the fundamental pulses (800 nm, 25 fs, 3k Hz) derived from the Ti: sapphire laser system with a 200-μm-thick beta barium borate (BBO) crystal. The fundamental pulse and second harmonic pulse aresynchronized in a Mach-Zehnder interferometer geometry. In each arm, the laser field intensity is controlled using a combination of a half-wave plate and a wire grid polarizer that are successively inserted in light path. The polarization of the fundamental pulse is turned to $z$ axis by rotating the wire grid polarizer, while the circular polarization of the second harmonic is guaranteed by rotating a quarter-wave plate placed after a wire grid polarizer in the path. The relative phase between the two colors is finely controlled by a pair of fused silica wedges mounted on a motorized stage. The pulses are then focused into a supersonic gas jet of Ar atoms in an ultrahigh vacuum chamber of COLTRIMS. A homogenous electric field of 3.2 V/cm and a magnetic field of 5.4 Gauss are exerted along $z$ axis (time-of-flight direction) in order to guide the charged fragments to the time- and position-sensitive detectors, from which the three-dimensional momentum distributions of electrons and ions can be reconstructed. Only the single ionization events, i.e., the electrons are coincident with $Ar^+$ ions, are presented. The intensity of 400 nm circular field is calibrated to be $1.42 \times 10^{14}$ W/cm$^2$ according to the location of ATI peaks as shown in Fig. 1b. The intensity of the weak 800 nm field is estimated by comparing the measured photoelectron energy spectrum in two-color fields with the solution of the time-dependent Schrödinger equation[48]. Atomic units (a.u.) are used throughout unless specified otherwise.

### Analytical derivation of the complex phase σ using saddle-point approach

The synthesized two-color light field can be expressed as $\mathbf{E}(t) = E_{2\omega}[\cos(2\omega t)\mathbf{z} + \sin(2\omega t)\mathbf{x}] + E_\omega \cos(\omega t + \varphi)\mathbf{z}$, with $E_{2\omega}$ and $E_\omega$ representing the electric field strengths of the two colors, and $\omega$ the laser frequency of the fundamental pulse. Using the expression of the electric field, we can derive the analytical formula for the complex phase $\sigma$, with its real part $\mathrm{Re}[\sigma]$ and imaginary part $\mathrm{Im}[\sigma]$ following

$$\mathrm{Re}[\sigma] = -\frac{p_z E_{800}}{\omega^2}\cos(\omega t_r + \varphi)\cosh(\omega t_i) + \frac{E_{400}E_{800}}{12\omega^3}\sin(3\omega t_r + \phi)\cosh(3\omega t_i)$$
$$- \frac{E_{400}E_{800}}{4\omega^3}\sin(\omega t_r - \phi)\cosh(\omega t_i)$$

(4a)

$$\mathrm{Im}[\sigma] = \frac{p_z E_{800}}{\omega^2}\sin(\omega t_r + \varphi)\sinh(\omega t_i) - \frac{E_{400}E_{800}}{12\omega^3}\cos(3\omega t_r + \phi)\sinh(3\omega t_i)$$
$$+ \frac{E_{400}E_{800}}{4\omega^3}\cos(\omega t_r - \phi)\sinh(\omega t_i)$$

(4b)

Here, $t_r$ and $t_i$ correspond to the real part and imaginary part of the saddle-point $t_s$. In Supplementary Fig. 1, we present the derived $t_s$ for electrons emitted within $[0, T_{400}]$ in the momentum space. Clearly, $t_r$ is mapped to the emission angle. $t_i$ reveals an isotropic property and it shows a pronounced dependence on the radial momentum $p_r$, i.e., with the increase of $p_r$, $t_i$ decreases. Based on the analytical formulas, we can directly calculate the amplitude and phase

modulations of the ionized electron wave function for the parallel ($\theta = 90°$) and perpendicular ($\theta = 0°$) interaction configurations. The corresponding results are shown in Supplementary Fig. 4, which agree well with the retrieved results from the SFA calculations (as displayed in Figs. 3a, b and 4a, b). Therefore, the validity of the Fourier transform analysis is confirmed.

## Data availability

The data supporting the findings of this study are available within the paper and its Supplementary Information files. The data generated in this study are provided in Supplementary Information/Source data file. Source data are provided with this paper.

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

## Acknowledgements

We thank the support of the National Key Research and Development Program of China (Grant Nos. 2022YFA1604301, Y.L. and 2023YFA1406800, P.G.) and National Science Foundation of China (Grant Nos. 12334013,92050201, 92250306, Y.L. and 8200906472, P.G.) and China Postdoctoral Science Foundation (No. 8206300495, P.G.).

## Author contributions

P.G. conceived the idea. P.G. and Y. Dou performed the experiment. P.G., M.H., and Y.L. analyzed the experimental data. P.G. and Y.F. performed the theoretical simulation. Y. Deng supported the operation of laser system. C.W. and Q.G. assisted the data analysis. This project was supervised by Y.L. All authors discussed the results and contributed to the final manuscript.

## Competing interests

The authors declare no competing interests.
