## [Peer Review File · Nature Communications]

Spatiotemporal imaging and shaping of electron wave functions using novel attoclock interferometryReviewer #1 (Remarks to the Author):

The manuscript titled "Spatiotemporal Imaging and Shaping of Electron Wave Functions Using Novel Attoclock Interferometry" demonstrates a novel attoclock technique to reconstruct the spatiotemporal evolution of the electron wave function from laser-induced tunneling ionization. In the experiments, using relative phase-controlled two-color laser fields, the authors measured three-dimensional momentum distributions of electrons from the single ionization of argon with a COLTRIMS apparatus. The experiment was well-designed, and the authors presented data with high quality. The interpretation of the momentum distributions with the help of SFA (Strong-Field Approximation) is convincing. The authors further analyze the measured momentum/energy distribution and retrieve the amplitude and the phase introduced by the weak 800 nm fields. However, I have my doubts about the analysis of the phase, and I have two major concerns about the results.

My first major concern is about the phase term (Volkov phase) used in SFA analysis of the photoelectron interference. The authors treated the Volkov phase as a complex phase and further used the real and imaginary parts of the complex phase to interpret the real phase and the amplitude induced by the 800 nm field. In the expression of the phase term, the momentum is a real operator, and the external field is a real function as well, which leads to the phase term being purely real. The authors should explain why the phase term becomes complex. Additionally, the authors didn't discuss the SFA amplitude term, which has also contribution to the wave function amplitude modulation due to the weak 800 nm field.

My second major concern is about the Coulomb effect. In the comparison between the momentum distributions from the experiment and SFA calculations, the authors pointed out the discrepancy between them, especially in the low-energy region. This discrepancy is not minor in terms of Fourier transform. It will lead to different amplitude values for a particular frequency term, which are used for further obtaining the real and imaginary parts of the "complex phase." A Coulomb-corrected SFA could probably help in this situation.

I have one minor concern about the phase term introduced by the ionization potential. I think it shall not be inside the integration over time. It shall be the phase inherited from the bound state at the time of ionization, $I_p \cdot t_s$. This should not change the calculations, but in my opinion, $I_p \cdot t_s$ is a more correct term.

Reviewer #2 (Remarks to the Author):

The paper presents a theoretical and experimental investigation of the manipulation of the electron wave packet created in tunneling ionization, employing a specific manipulation of the tunneling barrier. The tunneling barrier is manipulated in the longitudinal, as well as in the transverse directions. The full phase of the photoelectron momentum wave packet is retrieved experimentally based on the SFA analytical analysis.

Generally, I can underline the high quality of the investigation. The setup for the manipulation of the ionization barrier is unusual. The method of retrieval of the wave packet phase is astute. The paper shows a way for the manipulation of the amplitude and phase of the photoelectron wave packet and their measurement in an experiment. The results could be quite interesting for the specialists in strong-field physics.

However, for the general reader, the paper does not answer the question for which purpose the created wave packet can be employed, or what kind of valuable information on attosecond dynamics can be retrieved by this method, for instance, on charge migration and charge transfer processes in molecular ionization or other processes, as mentioned in the abstract, introduction, and conclusion of the paper. For this reason, I cannot recommend this manuscript for publication in NCOMMS.

Reviewer #3 (Remarks to the Author):

In the article of P. Ge et al, the authors, using two-color field, consisting of strong circularly polarized second harmonic and weak linearly polarized fundamental harmonics, aim to reconstruct the time-resolved evolution of electronic wavefunction. The idea behind the method is that by introducing the long-wavelength linear polarized field, the authors can modify the neighboring ionization instances, leading to interferences in frequency space. Analysis of these interferences, as the authors claim, allows to reconstruct the spatio-temporal evolution of the electronic wave function.

Although I find the results interesting and potentially scientifically sound, I'm not yet sure that the authors show what they claim to show, and that the article achieves the level of novelty devoting publication at the high level Nature Communications. Below I list my doubts in a hope that they can be resolved by the authors. Besides, I make several more technical remarks which hopefully help to improve the manuscript.

1) I see quite noticeable similarity to the previous work of the group: the previous work PRL 120, 073202 (2018). I agree, the pump configuration is different, but can it be that both pump configuration (here and in the earlier paper) produce similar, if not the same effect? If I see in the older paper, I see also that two fields 400 and 800 nm have different polarization directions at different ionization instances, which introduce actual difference in dynamics and thus difference in phase and interference. What I see that even the polarization diagrams in the older paper and in the present paper look the same. And the corresponding equations for the interference look for me also quite identical.

2) In the article I do not yet see the evolution of the electronic wavefunction in time. Here I see several missing pieces.

2a) First, only σ , the (complex) phase, resulting from the action of the weak 800 nm phase, is presented. But σ is only a part of the full phase. Probably, even the smallest part because the corresponding driving field is weak. Another term, for instance, is S_0 (defined on line 200). So, my question is, to which extent one can speak about reconstruction of the wavefunction if only σ is reconstructed?

b) The authors reconstruct spectrum in dependence on the angle. If one says, that the dynamics (behaviour in time) is reconstructed, there must be a mapping between time and angle. This mapping is influenced by two effects:

A) First, if the field is elliptically polarized, the mapping time-angle is distorted. The same happens if the field is not a single-color field, but has two colors. An example of such effects are shown for instance in I. Babushkin et al, Nat. Phys. 18, 417 (2022). I personally would estimate the "effective ellipticity" to be around 0.9, based on the given intensities, meaning that the "distortion" of the time-angle mapping is around $0.1 \cdot \text{cycle}$, which is not actually too small (but also not dramatically big).

B) The second effect comes from the Coulomb effect. Coulomb potential

deflects the wavepackets, making leading to an correction to the angle. As some people claim (for instance Torlina et al, Nat. Phys. 11, 503 (2015)), this deflection is solely responsible for the effect observed in the attoclock. The effect from this deflection must be actually of the same order, ~ 0.1 *cycle. Also not too big and but not too small.

The authors should at least acknowledge and discuss existence of such deflections but the best, of course, introduce corresponding corrections.

4) In view of said in the points 2 and 3, the following sentences sound for me at the moment a clear overstatement:

4a) "...we are able to reconstruct the spatio-temporal evolution of the electron wave function in momentum space within the optical cycle, including its amplitude and phase..."

4b) "... we fully reconstruct the temporal evolution of the electron wave function, thereby visualizing how the ionization dynamics evolves with respect to a fast-oscillating potential barrier ... "

4c) "The emission angle indeed encodes the spatial property of the field-modulated potential barrier."

... and some others of this type.

5) Taking into account said before, the authors should solidify their claim that the phase sigma they observe is the result of under-the-barrier dynamics (the author do claim this, or?). Whereas the dynamics of the imaginary part of sigma is relatively convincingly explained, the details of real part look still partially mysterious. How are they could be related to the "tunneling time" or similar quantities?

Finally, several more technical, minor remarks:

1) In Eq. 2, it is not clear for me, how the authors go from line 2 to the rest of the equation. For instance, for me it is not clear how the term like $3 * E_k * T_{400}$ can appear at all (the initial expression has the phase $\sim E_k$, but then I rise it to the power 2, so I expect the maximal factor 2 and not 3. Or I missed something? In any case, somewhat more detailed explanation of the derivation of Eq. 2 and its meaning would be useful (may be in Supplementary). For me this equation looks somewhat counter-intuitive.

2) What is the influence of the pulse envelope? Different intensities of different ionization instances lead to different conditions and also to different ionization patterns. Please make a remark on this.

3) At some point the authors say that if the amplitude of the perturbative field at 800 nm increases, the corresponding electron energy momentum decreases. Should it not be other way around?

4) In the sentence "We then add a weak linearly polarized field at 800 nm with the intensity of 8.8×10^{11} W/cm² along z direction" it is not clear what happens "along z direction" - polarization or propagation. Do I understand correctly, both input beams propagate along y direction, and polarization is consequently in x-z plane? I do not see

this said explicitly...

5) In the sentence "Here, the emission angle is defined as the angle between the direction of the electron final momentum p and $+z$ axis" do I understand correctly, the angle θ is mentioned?

6) I would recommend to write, instead of, say U^{800} $U^{(800)}$, and the same for other quantities, in order to avoid confusion of a superscript with mathematical operation of raising into power.

7) Some quantities such as T , T_{400} , T_{800} are seemingly not defined in the text.

8) The sentence "Varying the relative phase between two colors, the field configuration remains unchanged, whereas the perturbative field strength at the ionization time changes" is unclear. We have changed the field, what means then "field configuration unchanged"?

9) Fig. 2: the exact meaning of arrows in the upper row is not clear (we have one of the field of circular polarization, so I would naively expect a cycle rather than an arrow).

Reply to Reviewer #1

We thank Reviewer #1 very much for his/her constructive comments. We appreciate the reviewer for mentioning that “*the experiment was well-designed and the authors presented data with high quality*” and “*the interpretation of the momentum distributions with the help of SFA is convincing*”. While in the former manuscript, the reviewer has doubt about the analysis of the phase and two concerns about the results. In the following reply, we will address all the comments by Reviewer #1 in detail. We hope this reply and the revised manuscript could dispel Reviewer #1’s doubts and concerns.

#My first major concern is about the phase term (Volkov phase) used in SFA analysis of the photoelectron interference. The authors treated the Volkov phase as a complex phase and further used the real and imaginary parts of the complex phase to interpret the real phase and the amplitude induced by the 800 nm field. In the expression of the phase term, the momentum is a real operator, and the external field is a real function as well, which leads to the phase term being purely real. The authors should explain why the phase term becomes complex. Additionally, the authors didn’t discuss the SFA amplitude term, which has also contribution to the wave function amplitude modulation due to the weak 800 nm field.

Reply: Thanks for the reviewer’s comments. Reviewer #1 was concerned about the phase term (Volkov phase) used in SFA analysis of the photoelectron interference. As presented in the manuscript, the Volkov phase was expressed as $S(\mathbf{p}, t_s) = -\int_{t_s}^{\infty} [\mathbf{p} + \mathbf{A}(t)]^2 / 2 + I_p dt$. While the momentum \mathbf{p} is a real operator, the ionization instant t_s that is derived from the saddle-point equation (i.e., $[\mathbf{p} + \mathbf{A}(t_s)]^2 / 2 + I_p = 0$) is a complex number. This is because in the case of the atomic bound-states, the ionization potential satisfies $I_p > 0$, causing the complex solutions of t_s . When substituting the complex t_s into the expression of $S(\mathbf{p}, t_s)$, one would obtain a complex phase.

The reviewer was right that the SFA amplitude term ($\rho_s(\mathbf{p}) \sim \langle \mathbf{p} + \mathbf{A}(t_s) | \mathbf{r} \cdot \mathbf{E}(t_s) | \psi_0(\mathbf{r}) \rangle$) did also have contribution to the wave function amplitude modulation as the electric field $\mathbf{E}(t)$ and the vector potential $\mathbf{A}(t)$ are modified when introducing a weak 800 nm field. However, this modulation is much smaller as compared with the amplitude modulation induced by the change of imaginary part of the complex phase since the latter acts in an exponential form, i.e., $e^{-\text{Im}[\sigma]}$. To eliminate the concern of the reviewer about this point, we take the case in parallel interaction configuration as an example and directly calculate the amplitude term $\rho_s(\mathbf{p})$ in single- and two-color cases, which are labelled as $\rho_s^0(\mathbf{p})$ and $\rho_s^1(\mathbf{p})$ respectively. Then, we characterize the amplitude modulation induced by the amplitude term using the formula

of $\delta = \left| \frac{|\rho_s^1(\mathbf{p})| - |\rho_s^0(\mathbf{p})|}{|\rho_s^0(\mathbf{p})|} \right|$. The result of δ is shown in Fig. R1(a). For comparison, we

also present the amplitude modulation induced by the change of imaginary part of the complex phase. Likewise, we characterize it by using $\delta = \left| \frac{e^{-\text{Im}[S]} - e^{-\text{Im}[S_0]}}{e^{-\text{Im}[S_0]}} \right| = |e^{-\text{Im}[\sigma]} - 1|$.

Here, S_0 and S represent the complex phase of electrons accumulated in single- and two-color fields. The corresponding result is shown in Fig. R1(b). One could see that the amplitude modulation induced by the SFA amplitude term exhibits similar phase and energy dependences as that induced by the change of imaginary part of the complex phase. However, the magnitude of amplitude modulation induced by the former is quite smaller as compared with that induced by the latter. Therefore, in the following analysis we can neglect the amplitude modulation contribution induced by the SFA amplitude term when adding a weak 800 nm field. We have mentioned this point in the revised manuscript (see Line 217-220) and presented the calculated results in Supplementary Fig. 3.

Fig. R1| Calculated amplitude modulation of electron wave packet when adding a weak linearly polarized 800 nm field for parallel interaction configuration. (a) Amplitude modulation induced by the SFA amplitude term. (b) Amplitude modulation induced by the change of the imaginary part of complex phase.

#My second major concern is about the Coulomb effect. In the comparison between the momentum distributions from the experiment and SFA calculations, the authors pointed out the discrepancy between them, especially in the low-energy region. This discrepancy is not minor in terms of Fourier transform. It will lead to different amplitude values for a particular frequency term, which are used for further obtaining the real and imaginary parts of the “complex phase.” A Coulomb-corrected SFA could probably help in this situation.

Reply: Thanks for the comments. It is undeniable that the Coulomb effect would influence the photoelectron momentum distribution and eventually influence the extraction of the real and imaginary parts of the complex phase, especially in the lower energy region. Here, to identify the influence of Coulomb effect, we perform the Coulomb-corrected SFA (CCSFA) calculation. In this calculation, the saddle-point approximation has been adopted in order to derive the complex ionization instants. Based on the complex ionization instants, we then calculate the amplitude, phase and

ionization exit of the initial electron wave packet. Afterwards, we propagate the electron wave packet in the combined laser and Coulomb field. The calculated two-color phase-resolved photoelectron energy spectra at the emission angles of $\theta=0^\circ$, 45° , 90° and 135° using CCSFA model are shown in Fig. R2(i-l). For comparison, we also present the measured results in Fig. R2(a-d) together with the SFA calculated results as shown in Fig. R2(e-h). It is clearly visible that, influenced by the Coulomb potential, the yield of low-energy electrons in the energy spectra increases, indicating that the radial momentum of electron wave packet is decreased due to the Coulomb attraction. Moreover, the phase-resolved interference fringes in low-energy region slightly shift as compared with the SFA calculations and they show good agreement with the experimental results. This directly verifies that the discrepancy between experiment and SFA calculation as shown in Fig. 2 in the main text does arise from the Coulomb effect during the classical propagation of electrons.

In order to extract the amplitude modulation, we perform the Fourier transform analysis on the photoelectron energy spectra and obtain the zero-frequency terms in single- and two-color fields. In the SFA analysis, as illustrated in the main text, the two zero-frequency terms correspond to $4W_0^2$ and $2W_0^2(e^{2\text{Im}[\sigma]}+e^{-2\text{Im}[\sigma]})$, respectively. When taking the Coulomb effect into account, as the reviewer mentioned, the amplitude of the zero-frequency terms in single- and two-color fields changes. In view of the weak laser intensity of 800 nm field, we assume that the Coulomb effects exerted on photoelectrons in single- and two-color cases are nearly identical. Then, we can correct the amplitude of zero-frequency terms by multiplying a coefficient α to account for the influence of Coulomb effect, i.e., $\alpha 4W_0^2$ and $\alpha 2W_0^2(e^{2\text{Im}[\sigma]}+e^{-2\text{Im}[\sigma]})$. To obtain the amplitude modulation $\text{Im}[\sigma]$, we have to divide these two terms. As a result, α is eliminated. This suggests the Coulomb effect has little impact on the extraction of amplitude modulation.

As for the phase extraction, we have to admit that the Coulomb effect indeed influences the phase of electron wave packet. However, this influence is mainly confined within the low-energy region as shown in Fig. R2. Consequently, the extracted phase modulation from the measured spectra in the low-energy region would slightly deviate from the SFA calculated result. In spite of this defect, one can still learn the overall evolution of the electron wave packet with respect to the fast-oscillating 800 nm field from the retrieved amplitude and phase modulations based on SFA analysis. Importantly, one can notice that, at the emission angles of $\theta=0^\circ$ and 90° , which correspond to the perpendicular and parallel interaction configurations respectively, the interference patterns calculated by the SFA model [as shown in Fig. R2(e, g)] are basically consistent with that calculated using CCSFA model [as shown in Fig. R2(i, k)] without considering the amplitude modulation induced by the Coulomb attraction. This implies that in these two specific cases the Coulomb effect has tiny impact on the phase of electron wave packet and thus can be neglected in the following extraction.

Following the reviewer's suggestion, we have supplemented the CCSFA calculations in Supplementary Fig. 2 in order to clarify the Coulomb effect.

Fig. R2| Two-color phase-resolved photoelectron energy spectra at different emission angles. (a-d) Experimental results. (e-h) SFA calculations. (i-l) CCSFA calculations.

*#I have one minor concern about the phase term introduced by the ionization potential. I think it shall not be inside the integration over time. It shall be the phase inherited from the bound state at the time of ionization, $I_p * t_s$. This should not change the calculations, but in my opinion, $I_p * t_s$ is a more correct term.*

Reply: Thanks for the reviewer's comment. The reviewer was right that the phase term introduced by the ionization potential corresponds to the phase inherited from the bound state at the time of ionization and it should not be inside the integration over time. According to the suggestion of reviewer, we have modified this phase term with $I_p * t_s$ in the revised manuscript (see Line 169, 207 and 208).

Reply to Reviewer #2

We thank Reviewer #2 very much for his/her positive comments. We appreciate Reviewer#2 for thinking our investigation is “*of high quality*”. We are also grateful to the reviewer for mentioning that “*the setup for manipulation of the ionization barrier is unusual*” and “*the retrieval of the wave packet phase is astute*”. In particular, the reviewer thought “*the results could be quite interesting for the specialists in strong-field physics*”. We thank Reviewer #2 again for all these positive comments. Whereas the reviewer has a main concern about “*for which purpose the created wave packet can be employed or what kind of valuable information on attosecond dynamics can be retrieved by this method, for instance, on charge migration and charge transfer processes in molecular ionization or other processes, as mentioned in the abstract, introduction, and conclusion of the paper.*”. In the following, we will address the issue concerned by the reviewer.

In this work, we propose a novel two-color attoclock interferometry that enables spatiotemporal shaping and imaging of the ionized electron wave function. This field geometry provides us an ingenious way to spatiotemporally manipulate the potential barrier and to shape the ionized electron wave packet. The well-characterized electron wave packet in momentum space on sub-cycle scale might have potential to serve as the electron sources in generating ultrafast electron pluses in free space for time-resolved electron microscopy and diffraction [see P. Baum, Chem. Phys. 423, 55-61(2013), Y. Morimoto and P. Baum, Nat. Phys. 14, 252-256 (2018)]. Within that, one can real-time image the chemical reactions, surface dynamics and structural phase transformations. While in our present work, we regard the shaped electron wave packet as the indicator, from which we unravel the ultrafast ionization dynamics. For example, from the retrieved two-color phase (time)-resolved amplitude and phase modulation of electron wave packet at a specific angle as shown in Fig. 3 or Fig. 4, we could visualize how the ionization dynamics evolves with respect to the fast-oscillating potential barrier within an optical cycle for a specific interaction configuration. Moreover, from the retrieved angle-resolved electron wave packet in full momentum space as shown in Fig. 5, we are able to reveal the effect of spatial property of the potential barrier on the ionization dynamics. Particularly, as the amplitude of electron wave packet is tightly determined by the imaginary phase of electrons accumulated during the under-barrier motion, from the retrieved amplitude modulation one could also catch a glimpse of ultrafast electron dynamics in the classically forbidden region, i.e., under the field-modulated barrier, and thus revealing quantum nature of strong-field interaction. Our work illuminates puzzles in understanding how the spatiotemporal properties of Coulomb barrier influence the ionization process and shape the electron wave function. Detailed illustrations for attosecond ionization dynamics retrieved from the created electron wave packet can be found in the sections of “*Temporal shaping and imaging of electron wave function*” and “*Spatial shaping and imaging of electron wave function*” in the main text.

When it comes to the case of molecules, the ionization dynamics becomes much more complex. On the one hand, the molecular orbital would influence the amplitude

and phase of the ionized electron wave function [see D. Pavičić *et al.*, Phys. Rev. Lett. 98, 243001 (2007) and M. Meckel *et al.*, Nat. Phys. 10, 594 (2014)]. On the other hand, the anisotropic molecular potential would influence the following electron's excursion [S. Biswas *et al.*, Nat. Phys. 16, 778-783(2020)]. This means the ionized electron wave function carries the information of molecular orbital and the molecular potential. Completely characterizing the electron wave function of molecular photoionization using the proposed two-color attoclock approach would allow us to reveal the effects of molecular orbital and molecular potential. Particularly, equipped with the spatial imaging capability, the two-color attoclock approach might have important implications for probing the anisotropic molecular environment. Apart from these, it is known that the emission of photoelectrons would in the meantime initiate the ultrafast electronic dynamics inside molecules after photoionization. Shaping the emitted electron wave packet also leads to the modulation of electronic dynamics inside molecules. Since the spatial density of electrons inside molecules dictates the following charge migration and charge transfer processes, we could thus achieve ultrafast control and imaging of charge migration and charge transfer processes inside molecules by using the novel attoclock interferometry, paving the way to the real-time measurement and control of chemical reactions. Recent example that employs two-color fields to control charge transfer and chemical reaction can be seen in the work of L. Zhou *et al.* [see Nat. Chem. 15, 1229-1235 (2023)].

Reply to Reviewer #3

We thank Reviewer #3 very much for his/her careful and professional review of our work. We thank the reviewer for thinking our results are “*interesting and potentially scientifically sound*”. In particular, we are grateful to Reviewer #3 for pointing out the shortcomings and deficiencies of our work thereby helping us to improve it. All the comments and suggestions are very pertinent. According to the reviewer's comments and suggestions, we have modified the manuscript and Supplementary material correspondingly. In the following, we will address the comments given by Reviewer#3 point-by-point. We hope this reply and the revised manuscript could dispel the doubts of Reviewer #3 and convince him/her to give a recommendation for publication.

1) I see quite noticeable similarity to the previous work of the group: the previous work PRL 120, 073202 (2018). I agree, the pump configuration is different, but can it be that both pump configuration (here and in the earlier paper) produce similar, if not the same effect? If I see in the older paper, I see also that two fields 400 and 800 nm have different polarization directions at different ionization instances, which introduce actually difference in dynamics and thus difference in phase and interference. What I see that even the polarization diagrams in the older paper and in the present paper look the same. And the corresponding equations for the interference look for me also quite identical.

Reply: Thanks for the reviewer's comments. The reviewer has carefully reviewed the

present manuscript as well as our previous paper. He/she was right there were some similarities between the two works, such as the polarization configurations at different ionization instances and the interference equations. However, the two field geometries probe ionization dynamics in very different ways. In previous work [PRL 120, 073202 (2018)], we employed a weak corotating circular field at 800 nm to modulate the rotating potential barrier induced by 400 nm circular field and to probe the resultant ionization dynamics. In this field geometry, as illustrated in Fig. R3(a), different emission angles or different ionization instants correspond to different polarization configurations. Correspondingly, the rotating potential barrier is spatially manipulated by 800 nm field vector along different directions. Moreover, if one fixes the emission angle and varies the two-color relative phase, the two-color polarization configuration is also changing as the electric field vector of 800 nm circular field is rotating with respect to the phase. This corresponds to a scenario in which a potential barrier formed by the fixed 400 nm field vector is spatially modulated by the rotating 800 nm field vector. This means the angle-resolved and two-color phase-resolved results actually encode the same ionization dynamics, from which we can unravel how the spatially manipulated potential barrier under different polarization configurations shapes the electron wave function.

Whereas in the present work, we adopted a weak linearly polarized field at 800 nm to spatiotemporally manipulate the rotating potential barrier. In this case, the rotating potential barrier would experience the spatial manipulation along the polarization direction of 800 nm field. As depicted in Fig. R3(b), varying the emission angle, the polarization configuration of the two-color field vectors changes. Note that at the same time, the field strength of 800 nm field also changes. This is the first difference from the two-color corotating circular case. The second difference is that, when fixing the emission angle, the polarization configuration of the two field vectors is fixed. Varying the two-color relative phase is equivalent to changing the field strength of 800 nm probing field. Therefore, the two-color phase-resolved results at a specific angle actually encode the temporal evolution of the ionization dynamics under a fast-oscillating potential barrier.

In a word, the field geometry in previous work only probes ionization dynamics under different polarization configurations. It reveals how the spatial properties of Coulomb barrier influence the ionization dynamics and shape the electron wave function. While the field geometry employed in the present work probes more dynamical information during photoionization, apart from revealing the ionization dynamics under different polarization configurations, it also provides an ingenious way to unravel the temporal evolution of ionization dynamics through a fast-oscillating potential barrier at a specific interaction configuration, as illustrated in Fig. 3 and Fig. 4 in the main text.

Fig. R3| Angle(time)-dependent and two-color phase-dependent field configurations in two-color attoclock geometry. (a) Two-color field geometry consisting of a strong 400 nm circular field and a weak corotating 800 nm circular field. (b) Two-color field geometry consisting of a strong 400 nm circular field and a weak linearly polarized 800 nm field.

As for the interference pattern formula, it is universal for ionization by two-color ($\omega-2\omega$) multicycle laser fields that consist of a strong second harmonic field and a perturbative weak fundamental field. In such fields, the electron wave packets that are emitted in adjacent 2ω cycles are modulated by opposite fundamental ω field vector. And four electron wave packets that are released in consecutive two ω cycles can well account for the main features of interference pattern in two-color fields. However, one should note that by substituting the two-color synthesized electric fields into the interference pattern formula, different field geometry would lead to very distinct interference pattern.

#2) In the article I do not yet see the evolution of the electronic wavefunction in time. Here I see several missing pieces.

#2a) First, only sigma, the (complex) phase, resulting from the action of the weak 800 nm phase, is presented. But sigma is only a part of the full phase. Probably, even the smallest part because the corresponding driving field is weak. Another term, for instance, is S_0 (defined on line 200). So, my question is, to which extend one can speak about reconstruction of the wavefunction if only sigma is reconstructed?

Reply: Thanks for the reviewer's comments. The reviewer was right that in the former manuscript we only reconstructed σ instead of the full phase. The full phase should also include the phase term S_0 which is solely induced by the 400 nm circular fields. While the real part of S_0 corresponds to the phase of unperturbed electron wave function, the

imaginary part of S_0 is associated with the amplitude of unperturbed electron wave function. According to Eq. (3) in the main text, we could see the amplitude of the unperturbed electron wave function is encoded in the zero-frequency term of the interference pattern in single 400 nm fields, i.e., $4W_0^2$. Since $W_0 \sim e^{-\text{Im}[S_0]}$, one might be able to extract the imaginary part of S_0 . Whereas for the real part of S_0 , it is lost in interference pattern since the interference pattern records the phase difference between interfering electron wave packets rather than their phase. We admit that we cannot fully reconstruct the phase term of S_0 using the present method. But this defect does not hinder us to understand the temporal evolution of shaped electron wave function. Because from the definition of S_0 , i.e., $S_0(\mathbf{p}, t_s) = -\int_{t_s}^{\infty} [\mathbf{p} + \mathbf{A}_{400}(t)]^2 / 2 + I_p dt$, one could see that S_0 does not depend on the two-color relative phase. It behaves as a constant phase that depends on the photoelectron energy and ionization instant (or emission angle). Consequently, the retrieved two-color phase-dependent σ itself is sufficient to depict the temporal evolution of shaped electron wave function in two-color fields. Since the reviewer thought the statement about reconstruction of the electron wave packet was less rigorous, in the revised manuscript, we have modified it.

#b) The authors reconstruct spectrum in dependence on the angle. If one says, that the dynamics (behaviour in time) is reconstructed, there must be a mapping between time and angle. This mapping is influenced by two effects:

*A) First, if the field is elliptically polarized, the mapping time-angle is distorted. The same happens if the field is not a single-color field, but has two colors. An example of such effects is shown for instance in I. Babushkin et al, Nat. Phys. 18, 417 (2022). I personally would estimate the "effective ellipticity" to be around 0.9, based on the given intensities, meaning that the "distortion" of the time-angle mapping is around 0.1*cycle, which is not actually too small (but also not dramatically big).*

*B) The second effect comes from the Coulomb effect. Coulomb potential deflects the wavepackets, making leading to an correction to the angle. As some people claim (for instance Torlina et al, Nat. Phys. 11, 503 (2015)), this deflection is solely responsible for the effect observed in the attoclock. The effect from this deflection must be actually of the same order, ~0.1*cycle. Also, not too big and but not too small.*

The authors should at least acknowledge and discuss existence of such deflections but the best, of course, introduce corresponding corrections.

Reply: Thanks for the reviewer's comments. We agree with the reviewer that if one employs the attoclock technique to reconstruct the ionization dynamics in time, the time-to-angle mapping relationship should be carefully calibrated since it is susceptible to the field ellipticity and the Coulomb effect. As the reviewer mentioned, when adding a weak 800 nm linearly polarized field to the 400 nm circular field, the linear time-to-angle mapping in circular fields would be distorted. To quantify the distortion, we

directly calculate the time-to-angle relationship in the employed two-color fields using the SFA model. The result for two-color fields with the relative phase $\varphi=0$ is shown in Fig. R4 (a). For clear illustration, we select three cuts of the spectrum with the photoelectron radial momenta at 0.1a.u., 0.5a.u. and 1a.u. and display them in Fig. R4(b). As can be seen, after adding a weak 800 nm linearly polarized field, the linear time-to-angle mapping in 400 nm circular field is distorted. And the distortion shows a dependence on photoelectron energy. Increasing the photoelectron energy, the distortion decreases and the mapping relationship gradually becomes linear. However, the magnitude of the distortion is much smaller than that estimated by the reviewer based on the “effective ellipticity” of the two-color fields. This implies that simply treating the two-color synthesized fields as an elliptical field might be inappropriate. To clarify this point, we calculate the time-to-angle mapping in 400 nm elliptically polarized field with the ellipticity of 0.9. The result is shown in Fig. R4(c). Likewise, we present three cuts with p_r at 0.1 a.u., 0.5a.u. and 1 a.u. in Fig. R4(d). One can see that, the mapping relationship exhibits obvious nonlinear property, especially for low-energy region. In contrast to the case in two-color synthesized fields, the distortion of the time-to-angle mapping in elliptical fields is much larger. And its magnitude, especially in low-energy region, does approach the value estimated by the reviewer. This time-to-angle mapping in elliptical fields has been experimentally revealed in our previous work [see M. Han *et al.*, Nat. Photon.15, 765-771(2021)]. Our results demonstrate the distortion of the time-to-angle mapping in two-color synthesized fields does exist, but its magnitude is quite small and cannot be estimated using the picture of “effective ellipticity”.

Fig. R4| Calculated time-to-angle mapping relationship using SFA model. (a) Momentum-resolved time-to-angle mapping in two-color fields with $\varphi=0$. (b) Cuts of

(a) with the photoelectron radial momenta at 0.1a.u., 0.5a.u. and 1a.u.. (c) Momentum-resolved time-to-angle mapping in 400 nm elliptical fields with the ellipticity at 0.9. (d) Cuts of (c) with the photoelectron radial momenta at 0.1a.u., 0.5a.u. and 1a.u..

As for the second effect, i.e., Coulomb effect, we do admit that it has impacts on the electron wave packets. Particularly, as the reviewer mentioned, it would deflect the electron wave packet with a certain angle in momentum space, leading to the distortion of time-to-angle mapping in attoclock geometry. Here, to identify the distortion induced by Coulomb effect, we perform calculations using SFA model and Coulomb-corrected SFA (CCSFA) model. For clearly visualizing the deflection of electron wave packet, we have neglected the interference effect. The calculated angle-resolved photoelectron momentum spectra for two-color fields with $\varphi=0$ are shown Fig. R5. By comparison, one could see that, influenced by Coulomb potential, the radial momentum (energy) of electron wave packet decreases, and at the same time the electron wave packet is entirely rotated by a specific angle. This indicates the time-to-angle mapping also rotates angularly. To eliminate the Coulomb effect, one can rotate the experimental spectra with a specific angle. This angle can be obtained by comparing SFA calculations with CCSFA calculations. Actually, during the retrieval of the amplitude and phase modulations of electron wave packet in momentum space, as shown in Fig. 5 in the main text, we did rotate the experimental spectra in order to deduct the Coulomb effect. But in the former manuscript, we omitted to mention this point. We feel sorry for this. In the revised manuscript, we have added illustrations for it.

Fig. R5| Calculated angle-resolved photoelectron momentum (energy) spectra for two-color fields with $\varphi=0$. (a) SFA calculation. (b) CCSFA calculation.

We thank the reviewer again for the constructive comments and suggestions. Since the distortion of time-to-angle mapping induced by the addition of the weak 800 nm linearly polarized field is quite small [as shown in Fig. R4(a, b)], we can thus neglect it during the following analysis. As for the distortion induced by the Coulomb effect, we can correct it by rotating the experimental spectra with a specific angle. In the revised manuscript, we have acknowledged the two effects and added corresponding illustrations for them (see Line 195-196 and Line 364-366 in revised manuscript).

At last, we would like to emphasize that the reconstruction of the temporal

evolution of electron wave function is not based on the time-to-angle mapping of attoclock. In fact, the time-to-angle mapping was just used to determine the two-color polarization configurations at different emission angles. Instead, we achieve the time resolution by monitoring the two-color relative phase. The two-color phase-dependent results as shown in Fig. 3 and Fig. 4 in the main text actually reflect the temporal shaping of ionized electron wave packet through a fast-oscillating potential barrier within one 400 nm cycle.

4) *In view of said in the points 2 and 3, the following sentences sound for me at the moment a clear overstatement:*

4a) *"...we are able to reconstruct the spatio-temporal evolution of the electron wave function in momentum space within the optical cycle, including its amplitude and phase..."*

4b) *"... we fully reconstruct the temporal evolution of the electron wave function, thereby visualizing how the ionization dynamics evolves with respect to a fast-oscillating potential barrier ... "*

4c) *"The emission angle indeed encodes the spatial property of the field-modulated potential barrier." ... and some others of this type.*

Reply: The reviewer made these comments out of concern for points 2 and 3. In the preceding paragraphs, we have addressed points 2 and 3 in detail. Since we only reconstructed the complex phase σ , i.e., the shaping of electron wave packet instead of the full phase of electron wave function, we accept the reviewer's criticism on the first two sentences. In the revised manuscript, we have modified these sentences in order to make the statements more rigorous. The two sentences are modified into "*we are able to reconstruct the spatio-temporal evolution of the shaping on the amplitude and phase of electron wave function in momentum space within the optical cycle,*" and "*we fully reconstruct the temporal shaping of electron wave function, thereby visualizing how the ionization dynamics evolves with respect to a fast-oscillating potential barrier*". As for the third sentence, here we want to explain it. As illustrated in Fig. 1c in the main text, different emission angle corresponds to different polarization configuration. Consequently, the potential barrier at different emission angle would experience distinct spatial modulation, i.e., its spatial property changes with respect to the emission angle. Therefore, we state that "*the emission angle indeed encodes the spatial property of the field-modulated potential barrier*".

5) *Taking into account said before, the authors should solidify their claim that the phase sigma they observe is the result of under-the-barrier dynamics (the author do claim this, or?). Whereas the dynamics of the imaginary part of sigma is relatively convincingly explained, the details of real part look still partially mysterious. How are they could be related to the "tunneling time" or similar quantities?*

Reply: Thank for the reviewer's insightful comments. Actually, we did not claim the

complex phase σ was the result of under-the-barrier dynamics. According to the definition of σ , i.e., $\sigma = -\int_{t_s}^{\infty} [\mathbf{p} + \mathbf{A}_{400}(t)] \cdot \mathbf{A}_{800}(t, \varphi) dt$, we see the integral time starts from t_s to the end of laser pulse. This means σ represents the additional phase accumulated during the entire ionization process. Within the saddle-point approach, one can divide σ into two parts, i.e., $\sigma = \sigma_1 + \sigma_2$, with $\sigma_1 = -\int_{t_s}^{t_r} [\mathbf{p} + \mathbf{A}_{400}(t)] \cdot \mathbf{A}_{800}(t, \varphi) dt$ representing the additional phase accumulated during the under-barrier motion and $\sigma_2 = -\int_{t_r}^{\infty} [\mathbf{p} + \mathbf{A}_{400}(t)] \cdot \mathbf{A}_{800}(t, \varphi) dt$ denoting the additional phase accumulated in the classical propagation. Here, t_r is the real part of t_s with $t_s = t_r + it_i$. Since t_s is a complex number, after integration, we would obtain a complex phase σ_1 . While for σ_2 , it turns out to be a real phase because all the involved quantities are real. This indicates the imaginary part of σ , i.e., $\text{Im}[\sigma]$, solely results from the under-barrier motion, whereas the real part of σ , i.e., $\text{Re}[\sigma]$, results from the entire ionization process. Therefore, by inspecting the spectra of $\text{Im}[\sigma]$, we can shed light on the under-barrier dynamics of electrons. However, for the spectra of $\text{Re}[\sigma]$, honestly speaking, it is hard to experimentally resolve the individual contributions from the under-barrier dynamics and subsequent classical propagation, thus making the encoded dynamics obscure. This might be the reason why the reviewer thought “*the details of real part look still partially mysterious*”.

Alternatively, we can understand the dynamics imprinted on the spectra of $\text{Re}[\sigma]$ with the help of SFA model. Here, we separately calculate the phase contributions of $\text{Re}[\sigma]$ in the under-barrier motion and in the subsequent propagation, which correspond to $\text{Re}[\sigma_1]$ and $\text{Re}[\sigma_2]$ (or σ_2), respectively. The calculated two-color phase-resolved $\text{Re}[\sigma_1]$ and $\text{Re}[\sigma_2]$ for parallel and perpendicular interaction configurations are shown in Fig. R6. Clearly, one could see that, in parallel polarization configuration, the phase contribution resulting from the under-barrier dynamics [Fig. R6(a)] is much larger than that accumulated in the classical propagation [see Fig. R6(b)]. This indicates in parallel polarization configuration, the retrieved $\text{Re}[\sigma]$ spectra as shown in Figs. 3(b) and 3(d) in the main text actually reflect the main features of the real phase modulation accumulated during under-barrier motion. With the help of the calculation shown in Fig. R6(a), we can understand the real phase modulation under the barrier as following: perturbed by the 800 nm field, the potential barrier is modified and this modification is determined by the instantaneous electron field $\mathbf{E}_{800}(t)$. At the same time, the electron would obtain an additional momentum along the ionizing direction that depends on the negative vector potential $-\mathbf{A}_{800}(t)$. In this case, when the 800 nm field vanishes, the potential barrier remains unchanged, whereas the momentum the electron obtains from the 800 nm field approaches maximum. This leads to the largest phase modulation on the real phase of electron during the under-barrier motion. When the 800 nm field approaches the maximum, the additional momentum imparted by the 800 nm field vanishes. Correspondingly, the modulation on the real phase of electron disappears. While the situation changes after the electron ionizes through the potential barrier, as shown in Fig. R6(b). Because the additional momentum acquiring from the 800 nm

field becomes perpendicular to the final momentum of electron. This indicates the trajectory of electron would experience a transverse perturbation during classical propagation. Along this trajectory, electron would obtain an additional transverse phase that depends on the momentum imparted by 800 nm field, as shown in Fig. R6(b). And this additional phase does not depend on the photoelectron energy.

As for the perpendicular polarization configuration, we could see the retrieved phase modulation $\text{Re}[\sigma]$ (as shown in Fig. 4(b) and 4(d) in the main text) mainly comes from the contribution of the classical propagation, i.e., $\text{Re}[\sigma_2]$, as shown in Fig. R6(d). In this configuration, the potential barrier experiences a transverse distortion that depends on $E_{800}(t)$. Meanwhile, the electron would obtain an additional transverse momentum perpendicular to the ionizing direction. Correspondingly, the electron trajectory in the barrier would be modified along the transverse direction. Based on the calculations shown in Fig. R6(c) and 6(d), we can understand the real phase modulation in perpendicular interaction configuration as follows. When the 800 nm field reaches the maximum, the potential barrier is transversely distorted to the utmost extent. Although the electron does not obtain an additional transverse momentum from 800 nm fields, the transverse excursion or displacement experiences the largest modulation, thus leading to the largest real phase modulation under the barrier as shown in Fig. R6(c). When the 800 nm field vanishes, the potential barrier remains unchanged. Correspondingly, the transverse displacement of electron under the barrier keeps unchanged. In this case, the real phase modulation during the under-barrier motion disappears [see Fig. R6(c)]. Since the transverse displacement at the ionization exit significantly influences the subsequent classical propagation, the electron would obtain the largest phase modulation during the classical propagation when the change of the transverse displacement at the ionization exit reaches the maximum, as shown in Fig. R6(d). When the change of the transverse displacement at the ionization exit vanishes, the real phase modulation disappears [see Fig. R6(d)].

In the revised Supplementary Information, we have added the calculated phase modulations of $\text{Re}[\sigma_1]$ and $\text{Re}[\sigma_2]$ in Supplementary Fig. 5 in order to better understand the electron dynamics imprinted in the retrieved phase modulation spectra.

Fig. R6| Calculated phase modulations resulting from the under-barrier dynamics and the subsequent classical propagation for parallel and perpendicular interaction configurations. (a) Phase modulation $\text{Re}[\sigma_1]$ in under-barrier motion for parallel configuration. (b) Phase modulation $\text{Re}[\sigma_2]$ accumulated in classical propagation for parallel configuration. (c) Phase modulation $\text{Re}[\sigma_1]$ in under-barrier motion for perpendicular configuration. (d) Phase modulation $\text{Re}[\sigma_2]$ accumulated in classical propagation for perpendicular configuration.

The reviewer cared about whether the retrieved amplitude or phase modulation can be related with the tunneling time or some other quantities. Actually, one might be able to reconstruct the imaginary part of the complex ionization instant (or saddle point), i.e., t_i , based on the retrieved amplitude modulation since this term corresponds to the imaginary phase of σ and is associated with t_i . In tunneling regime, t_i is generally interpreted as the Keldysh time, which quantifies the time the electron spends under the barrier. However, according to the employed laser parameters in our work, we determine the Keldysh parameter here is about 2.72. This demonstrates the ionization is located at nonadiabatic tunneling regime [G. L. Yudin and M. Y. Ivanov, Phys. Rev. A, 64, 013409 (2001)], where tunneling and multiphoton absorption coexist. Thus, t_i cannot be directly interpreted as the Keldysh time or tunneling time. It might be used to characterize the nonadiabaticity of photoionization in our work. As for the retrieved phase modulation, if calculating its momentum derivative, one might be able to link it with the offset of the ionization exit.

Finally, several more technical, minor remarks:

1) In Eq. 2, it is not clear for me, how the authors go from line 2 to the rest of the equation. For instance, for me it is not clear how the term like $3 * E_k * T_{400}$ can appear at all (the initial expression has the phase $\sim E_k$, but then I rise it to the power 2, so I expect the maximal factor 2 and not 3. Or I missed something? In any case, somewhat more detailed explanation of the derivation of Eq. 2 and its meaning would be useful (may be in Supplementary). For me this equation looks somewhat counter-intuitive.

Reply: Thanks for the reviewer's comments. It's our carelessness that in the former presentation we did not give a detailed derivation of the interference formula as expressed in Eq. 2 that makes the reviewer confused. In the following, we will show the derivation of Eq. 2 step by step. As illustrated in the main text, the interference pattern in two-color fields can be well accounted for by the interference of four electron wave packets released in two consecutive 800 nm cycles, i.e.,

$I(\mathbf{p}, \varphi) = |\psi_1 + \psi_2 + \psi_3 + \psi_4|^2$ as expressed in Eq.1. As the electron wave packet can be

described using $\psi = \rho(\mathbf{p})e^{iS}$, with $\rho_s(\mathbf{p}) \sim \langle \mathbf{p} + \mathbf{A}(t_s) | \mathbf{r} \cdot \mathbf{E}(t_s) | \psi_0(\mathbf{r}) \rangle$ denoting the pre-

exponential factor and $S(\mathbf{p}, t_s) = -\int_{t_s}^{\infty} [\mathbf{p} + \mathbf{A}_{400}(t) + \mathbf{A}_{800}(t, \varphi)]^2 / 2 + I_p dt$ representing

the complex phase of electrons in two-color fields, we can write Eq. 1 in the main text

as $I(\mathbf{p}, \varphi) = |\psi_1 + \psi_2 + \psi_3 + \psi_4|^2 = |\rho_{s1}(\mathbf{p})e^{iS_1} + \rho_{s2}(\mathbf{p})e^{iS_2} + \rho_{s3}(\mathbf{p})e^{iS_3} + \rho_{s4}(\mathbf{p})e^{iS_4}|^2$.

Considering the two-color field periodicity, the ionization instants of the four electron wave packets satisfy $t_{s3} = t_{s1} + T_{800}$ and $t_{s4} = t_{s2} + T_{800}$. With that, the two-color synthesized fields at the ionization instants satisfy $\mathbf{E}(t_{s3}) = \mathbf{E}(t_{s1})$ and $\mathbf{E}(t_{s4}) = \mathbf{E}(t_{s2})$. Correspondingly,

$\mathbf{A}(t_{s3}) = \mathbf{A}(t_{s1})$ and $\mathbf{A}(t_{s4}) = \mathbf{A}(t_{s2})$. As a result, $\rho_{s3}(\mathbf{p}) = \rho_{s1}(\mathbf{p})$ and $\rho_{s4}(\mathbf{p}) = \rho_{s2}(\mathbf{p})$. As

for the complex phase, it is governed by the following relationship:

$$\begin{aligned} S_3 &= -\int_{t_{s1}+T_{800}}^{\infty} [\mathbf{p} + \mathbf{A}_{400}(t) + \mathbf{A}_{800}(t, \varphi)]^2 / 2 + I_p dt \\ &= -\int_{t_{s1}}^{\infty} [\mathbf{p} + \mathbf{A}_{400}(t) + \mathbf{A}_{800}(t, \varphi)]^2 / 2 + I_p dt + \int_0^{T_{800}} [\mathbf{p} + \mathbf{A}_{400}(t) + \mathbf{A}_{800}(t, \varphi)]^2 / 2 + I_p dt \\ &= S_1 + b \end{aligned}$$

$$\begin{aligned} S_4 &= -\int_{t_{s2}+T_{800}}^{\infty} [\mathbf{p} + \mathbf{A}_{400}(t) + \mathbf{A}_{800}(t, \varphi)]^2 / 2 + I_p dt \\ &= -\int_{t_{s2}}^{\infty} [\mathbf{p} + \mathbf{A}_{400}(t) + \mathbf{A}_{800}(t, \varphi)]^2 / 2 + I_p dt + \int_0^{T_{800}} [\mathbf{p} + \mathbf{A}_{400}(t) + \mathbf{A}_{800}(t, \varphi)]^2 / 2 + I_p dt \\ &= S_2 + b \end{aligned}$$

Here, $b = \int_0^{T_{800}} [\mathbf{p} + \mathbf{A}_{400}(t) + \mathbf{A}_{800}(t, \varphi)]^2 / 2 + I_p dt$. It corresponds to the phase difference

due to the time difference of traveling in the continuum and can be reduced to a constant phase with $b = (U_p^{(400)} + U_p^{(800)} + E_k + I_p)T_{800}$. Here, $E_k = |\mathbf{p}|^2/2$ denotes the photoelectron

energy. U_p is the pondermotive energy, with $U_p^{(400)} = E_{400}^2 / (8\omega^2)$ and $U_p^{(800)} = E_{800}^2 / (4\omega^2)$. Then, the interference formula can be rewritten as:

$$\begin{aligned}
I(\mathbf{p}, \varphi) &= |\psi_1 + \psi_2 + \psi_3 + \psi_4|^2 \\
&= \left| \rho_{s1}(\mathbf{p})e^{iS_1} + \rho_{s2}(\mathbf{p})e^{iS_2} + \rho_{s1}(\mathbf{p})e^{iS_1+ib} + \rho_{s2}(\mathbf{p})e^{iS_2+ib} \right|^2 \\
&= \left| \rho_{s1}(\mathbf{p})e^{iS_1} + \rho_{s2}(\mathbf{p})e^{iS_2} \right|^2 |1 + e^{ib}|^2 \\
&= 2 \left| \rho_{s1}(\mathbf{p})e^{iS_1} + \rho_{s2}(\mathbf{p})e^{iS_2} \right|^2 [1 + \cos(b)]
\end{aligned} \tag{R(1)}$$

Since 800 nm is perturbative weak, we assume it has tiny effect on the ionization instants of the electron wave packets ψ_1 and ψ_2 and the pre-exponential factor $\rho_s(\mathbf{p})$. Therefore, $t_{s2} = t_{s1} + T_{400}$, and $\rho_{s1}(\mathbf{p}) \approx \rho_{s2}(\mathbf{p})$. In the following, we label $\rho_{s1}(\mathbf{p})$ and $\rho_{s2}(\mathbf{p})$ as $\rho_s(\mathbf{p})$. Correspondingly, the complex phase of the two electron wave packets (ψ_1 and ψ_2) can be expressed as

$$\begin{aligned}
S_1 &= -\int_{t_{s1}}^{\infty} [\mathbf{p} + \mathbf{A}_{400}(t) + \mathbf{A}_{800}(t, \varphi)]^2 / 2 + I_p dt \\
&= -\int_{t_{s1}}^{\infty} [\mathbf{p} + \mathbf{A}_{400}(t)]^2 / 2 + I_p dt - \int_{t_{s1}}^{\infty} [\mathbf{p} + \mathbf{A}_{400}(t)]\mathbf{A}_{800}(t, \varphi) dt - \int_{t_{s1}}^{\infty} [\mathbf{A}_{800}(t, \varphi)]^2 / 2 \\
&= S_{10} + \sigma_1 + \alpha_1 \\
S_2 &= -\int_{t_{s2}}^{\infty} [\mathbf{p} + \mathbf{A}_{400}(t) + \mathbf{A}_{800}(t, \varphi)]^2 / 2 + I_p dt \\
&= -\int_{t_{s2}}^{\infty} [\mathbf{p} + \mathbf{A}_{400}(t)]^2 / 2 + I_p dt - \int_{t_{s2}}^{\infty} [\mathbf{p} + \mathbf{A}_{400}(t)]\mathbf{A}_{800}(t, \varphi) dt - \int_{t_{s2}}^{\infty} [\mathbf{A}_{800}(t, \varphi)]^2 / 2 \\
&= S_{20} + \sigma_2 + \alpha_2 \\
&= S_{10} + (E_k + U_p^{400} + I_p)T_{400} + \sigma_2 + \alpha_1 + U_p^{800}T_{400} \\
&= S_{10} + \sigma_2 + \alpha_1 + b / 2
\end{aligned} \tag{R(2)}$$

Here, $S_0 = -\int_{t_s}^{\infty} [\mathbf{p} + \mathbf{A}_{400}(t)]^2 / 2 + I_p dt$ represents the phase induced solely by 400 nm circular fields. $\sigma = -\int_{t_s}^{\infty} [\mathbf{p} + \mathbf{A}_{400}(t)]\mathbf{A}_{800}(t, \varphi) dt$ is the additional phase induced by the weak 800 nm linearly polarized field. $\alpha = -\int_{t_s}^{\infty} [\mathbf{A}_{800}(t, \varphi)]^2 / 2 = U_p^{800} t_s$ is a high-order small quantity that depends on the laser intensity of 800 nm fields, therefore in the following derivation we can neglect its contribution. By substituting the phases in Eq. R(2) into Eq. R(1), one can obtain the following formula:

$$\begin{aligned}
I(\mathbf{p}, \varphi) &= |\psi_1 + \psi_2 + \psi_3 + \psi_4|^2 \\
&= 2 \left| \rho_s(\mathbf{p})e^{iS_1} + \rho_s(\mathbf{p})e^{iS_2} \right|^2 [1 + \cos(b)] \\
&= 2 \left| \rho_s(\mathbf{p})e^{iS_{10} + \sigma_1} + \rho_s(\mathbf{p})e^{iS_{10} + \sigma_2 + b/2} \right|^2 [1 + \cos(b)]
\end{aligned} \tag{R(3)}$$

Then, we use $W_0 = \rho(\mathbf{p})e^{-\text{Im}[S_{10}]}$ and $\text{Re}[S_{10}]$ to denote the amplitude and phase of the unperturbed electron wave function in 400 nm circular fields. Accordingly, the

interference formula can be arranged into:

$$\begin{aligned}
I(\mathbf{p}, \varphi) &= |\psi_1 + \psi_2 + \psi_3 + \psi_4|^2 \\
&= 2 \left| W_0 e^{i\text{Re}[S_{10}]} e^{i\sigma_1} + W_0 e^{i\text{Re}[S_{10}]} e^{i\sigma_2 + ib/2} \right|^2 [1 + \cos(b)]. \quad \text{R(4)} \\
&= 2W_0^2 \left| e^{i\sigma_1} + e^{i\sigma_2 + ib/2} \right|^2 [1 + \cos(b)]
\end{aligned}$$

From this formula, one can see that the phase information (i.e., $\text{Re}[S_{10}]$) of the unperturbed electron wave function is lost when considering the interference effect. Further, we deduce the analytical formula of σ according to its definition for two-color fields with $\mathbf{E}(t) = E_{2\omega}[\cos(2\omega t)\mathbf{z} + \sin(2\omega t)\mathbf{x}] + E_{\omega}\cos(\omega t + \varphi)\mathbf{z}$, which is expressed as

$$\begin{aligned}
\sigma(t_s) &= -\int_{t_s}^{\infty} [\mathbf{p} + \mathbf{A}_{400}(t)] \mathbf{A}_{800}(t, \varphi) dt \\
&= -\frac{p_z E_{800}}{\omega^2} \cos(\omega t_s + \varphi) + \frac{E_{400} E_{800}}{12\omega^3} \sin(3\omega t_s + \varphi) - \frac{E_{400} E_{800}}{4\omega^3} \sin(\omega t_s - \varphi) \quad \text{R(5)}
\end{aligned}$$

As $t_{s2} = t_{s1} + T_{400}$, we find $\sigma_2 = -\sigma_1$. This indicates that the two electron wave packets emitted from adjacent 400 nm cycles experience opposition amplitude and phase modulations. In the following, we denote σ_1 as $-\sigma$ and σ_2 as σ . Then, we rearrange the interference formula as:

$$\begin{aligned}
I(\mathbf{p}, \varphi) &= |\psi_1 + \psi_2 + \psi_3 + \psi_4|^2 \\
&= 2W_0^2 \left| e^{-i\sigma} + e^{i\sigma + ib/2} \right|^2 [1 + \cos(b)] \\
&= 2W_0^2 \left| e^{\text{Im}[\sigma]} e^{-i\text{Re}[\sigma]} + e^{-\text{Im}[\sigma]} e^{i\text{Re}[\sigma] + ib/2} \right|^2 [1 + \cos(b)] \quad \text{R(6)} \\
&= 2W_0^2 [e^{2\text{Im}[\sigma]} + e^{-2\text{Im}[\sigma]} + 2\cos(2\text{Re}[\sigma] + b/2)] [1 + \cos(b)] \\
&= 2W_0^2 [e^{2\text{Im}[\sigma]} + e^{-2\text{Im}[\sigma]}] [1 + \cos(b)] + 4W_0^2 \cos(2\text{Re}[\sigma] + b/2) \\
&\quad + 2W_0^2 \cos[2\text{Re}[\sigma] - b/2] + 2W_0^2 \cos[2\text{Re}[\sigma] + 3b/2]
\end{aligned}$$

Note that $b = (U_p^{(400)} + U_p^{(800)} + E_k + I_p)T_{800}$, we can further arrange it into the formula of $b = 2E_k T_{400} + 2a$, with $a = (U_p^{(400)} + U_p^{(800)} + I_p)T_{400}$. Therefore, the interference formula in the two-color fields can be finally expressed as

$$\begin{aligned}
I(E_k, \varphi) &= |\psi_1 + \psi_2 + \psi_3 + \psi_4|^2 \\
&= 2W_0^2 (e^{2\text{Im}[\sigma]} + e^{-2\text{Im}[\sigma]}) [1 + \cos(2E_k T_{400} + 2a)] \\
&\quad + 4W_0^2 \cos(E_k T_{400} + 2\text{Re}[\sigma] + a) \quad \text{R(7)} \\
&\quad + 2W_0^2 \cos(E_k T_{400} - 2\text{Re}[\sigma] + a) \\
&\quad + 2W_0^2 \cos(3E_k T_{400} + 2\text{Re}[\sigma] + 3a)
\end{aligned}$$

According to the reviewer's suggestion, in the revised Supplementary Information, we have added the detailed derivation for Eq. 2.

2) What is the influence of the pulse envelope? Different intensities of different ionization instances lead to different conditions and also to different ionization patterns. Please make a remark on this.

Reply: Thanks for the reviewer's comments. In the former manuscript, we did not

consider the effect of the pulse envelope. As the reviewer said, the pulse envelope would influence the amplitude and phase of the ionization bursts at different ionization instants. If the pulse duration is short or it is reduced to few-cycle timescale, the effect of the pulse envelope definitely plays an important role in modifying the amplitude and phase of the electron wave function, and thus the resultant interference pattern. This point can be verified by referring to the related works that employed the few-cycle laser pulses to trigger ionization (for example, see R. Gopal et al., PRL 103, 053001 (2009)). Whereas, if the pulse duration is longer, the effect of the pulse envelope on the electron wave packets and their interference becomes less important. This exactly corresponds to the case in our experiment as the pulse duration of the employed 400 nm circular fields is about 30 cycles. In this case, the amplitude and phase of electron wave packets emitted at different ionization instants would be slightly modified by the pulse envelope. As a result, the interference pattern changes slightly. While the reconstruction of electron wave packet is based on the measured cycle-integrated interference pattern, the retrieved electron wave function actually reflects the averaged result within multiple cycles.

3) At some point the authors say that if the amplitude of the perturbative field at 800 nm increases, the corresponding electron energy momentum decreases. Should it not be other way around?

Reply: Thanks for the reviewer's comments. Actually, this sentence specifically refers to the energies of the interference fringes, i.e., the ATI peaks and sideband structures. From the perspective of frequency domain, these interference fringes can be regarded as the result of multiphoton absorption. In the employed two-color fields, the energies of ATI peaks and sidebands are respectively governed by $E_k=2n\omega-I_p-U_p^{(400)}-U_p^{(800)}$ and $E_k=(2n\pm 1)\omega-I_p-U_p^{(400)}-U_p^{(800)}$. Here, n represents the number of absorbed photons from 400 nm circular fields. I_p is the ionization potential. U_p is pondermotive energy. Based on the formulas, one could see that increasing the field strength of 800 nm field, $U_p^{(800)}$ increases. This is equivalent to increasing the effective ionization potential. As a result, the energies of interference fringes decrease.

4) In the sentence "We then add a weak linearly polarized field at 800 nm with the intensity of 8.8×10^{11} W/cm² along z direction" it is not clear what happens "along z direction" - polarization or propagation. Do I understand correctly, both input beams propagate along y direction, and polarization is consequently in x-z plane? I do not see this said explicitly..

Reply: Thanks for the reviewer's comments. We are sorry for the ambiguous expressions in the former manuscript. The reviewer was right that both input beams propagated along y direction. The polarization of the weak 800 nm fields was along z direction, and the polarization plane of the intense 400 nm circular fields corresponded to x-z plane. Following the reviewer's comments, in the revised manuscript, we have modified the expression into "We then add a weak linearly polarized field at 800 nm with the intensity of 8.8×10^{11} W/cm² and its polarization along z direction" (see Line 104 in the main text). In addition, we have also added the expression that "Here, the

polarization plane of 400 nm circular fields is in x - z plane.” to clearly illustrate the polarization plane of 400 nm circular fields (see Line 89 in the main text).

5) In the sentence "Here, the emission angle is defined as the angle between the direction of the electron final momentum p and $+z$ axis" do I understand correctly, the angle θ is mentioned?

Reply: Yes, the emission angle defined here exactly refers to θ . It is our carelessness for omitting the label of the emission angle. In the revised manuscript, we have modified this sentence into "Here, the emission angle θ is defined as the angle between the direction of the electron final momentum p and $+z$ axis." (see Line 92 in the main text).

6) I would recommend to write, instead of, say U^{800} $U^{(800)}$, and the same for other quantities, in order to avoid confusion of a superscript with mathematical operation of raising into power.

Reply: Thanks for the reviewer's suggestion. Following the reviewer's suggestion, we have changed the form of the superscript of all the quantities. For example, we have changed U_p^{800} into $U_p^{(800)}$, and U_p^{400} into $U_p^{(400)}$.

7) Some quantities such as T , T_{400} , T_{800} are seemingly not defined in the text.

Reply: Thanks for the reviewer's reminding. In the revised manuscript, we have added the definitions of these quantities (see Line 176 and 178 in the main text).

8) The sentence "Varying the relative phase between two colors, the field configuration remains unchanged, whereas the perturbative field strength at the ionization time changes" is unclear. We have changed the field, what means then "field configuration unchanged"?

Reply: We are sorry for this mistake. Here, we intended to say "the polarization configuration" remains unchanged rather than the "field configuration". In the revised manuscript, we have corrected it.

9) Fig. 2: the exact meaning of arrows in the upper row is not clear (we have one of the fields of circular polarization, so I would naively expect a cycle rather than an arrow).

Reply: In Fig. 2, we used the blue arrow to denote the direction of the rotating electric field vector of 400 nm circular pulses at different ionization instant (or emission angle). The red arrow was used to indicate the polarization direction of 800 nm linearly polarized fields. Since the reviewer thought this representation was not clear, in the revised Fig. 2, we have added a blue dashed cycle to illustrate the circularly polarized feature of 400 nm fields.

Reviewer #1 (Remarks to the Author):

In their detailed replies, the authors addressed my concerns, along with those of the other two reviewers, in the previous review reports and revised the manuscript accordingly. The additional results with Coulomb-corrected SFA are impressive and convincing. Supplementary information, aiding the comprehension of phases discussed in the main text, has been added and is included in the Supplemental Material. I recommend the publication of this manuscript in Nature Communications.

Reviewer #2 (Remarks to the Author):

In their response, the authors discussed in every detail the comments of Reviewers #1 and #3, convincingly explaining their points and adding pertinent corrections to the paper text and in the Supplement. I have been also satisfied with the reasoning related to my comment. Consequently, I recommend the revised paper for publication in NCOMM.

Reviewer #3 (Remarks to the Author):

After careful consideration of the answers given by the authors to all the questions, I see that my criticisms have been carefully addressed. The authors have been able to convince me that the field configuration is indeed significantly different from that described in their previous articles. The authors have also carefully addressed the "Coulomb tail" problem (which was also raised by the other reviewers). To summarize, I'm now convinced that the article is scientifically sound, and has sufficient novelty and significance. I therefore recommend to publish it in Nature Communications as it is, without further changes.